# *TUBB4A* mutations result in both glial and neuronal degeneration in an H-ABC leukodystrophy mouse model

Sunetra Sase[1†], Akshata A Almad[1†], C Alexander Boecker[2], Pedro Guedes-Dias[2], Jian J Li[3], Asako Takanohashi[1], Akshilkumar Patel[1], Tara McCaffrey[1], Heta Patel[1], Divya Sirdeshpande[1], Julian Curiel[1], Judy Shih-Hwa Liu[4], Quasar Padiath[5], Erika LF Holzbaur[2], Steven S Scherer[3], Adeline Vanderver[1,3]*

[1]Department of Neurology, The Children's Hospital of Philadelphia, Philadelphia, United States; [2]Department of Physiology, the Perelman School of Medicine, University of Pennsylvania, Philadelphia, United States; [3]Department of Neurology, the Perelman School of Medicine, University of Pennsylvania, Philadelphia, United States; [4]Department of Neurology, Department of Molecular Biology, Cell Biology, and Biochemistry, Brown University, Providence, United States; [5]Department of Human Genetics and Neurobiology, University of Pittsburgh, Pittsburgh, United States

*For correspondence:
vandervera@email.chop.edu

[†]These authors contributed equally to this work

**Abstract** Mutations in *TUBB4A* result in a spectrum of leukodystrophy including Hypomyelination with Atrophy of Basal Ganglia and Cerebellum (H-ABC), a rare hypomyelinating leukodystrophy, often associated with a recurring variant p.Asp249Asn (D249N). We have developed a novel knock-in mouse model harboring heterozygous ($Tubb4a^{D249N/+}$) and the homozygous ($Tubb4a^{D249N/D249N}$) mutation that recapitulate the progressive motor dysfunction with tremor, dystonia and ataxia seen in H-ABC. $Tubb4a^{D249N/D249N}$ mice have myelination deficits along with dramatic decrease in mature oligodendrocytes and their progenitor cells. Additionally, a significant loss occurs in the cerebellar granular neurons and striatal neurons in $Tubb4a^{D249N/D249N}$ mice. In vitro studies show decreased survival and dysfunction in microtubule dynamics in neurons from $Tubb4a^{D249N/D249N}$ mice. Thus $Tubb4a^{D249N/D249N}$ mice demonstrate the complex cellular physiology of H-ABC, likely due to independent effects on oligodendrocytes, striatal neurons, and cerebellar granule cells in the context of altered microtubule dynamics, with profound neurodevelopmental deficits.

## Introduction

Hypomyelination with Atrophy of Basal Ganglia and Cerebellum (H-ABC) is a leukodystrophy caused by sporadic, typically de novo, heterozygous mutations in the *TUBB4A* gene (*Simons et al., 2013*). This gene encodes the tubulin beta 4A protein, which heterodimerizes with α−tubulin to form subunits that assemble into microtubules. Monoallelic mutations in *TUBB4A* result in a spectrum of neurologic disorders ranging from an early onset leukoencephalopathy to adult-onset Dystonia type 4 (DYT4; Whispering Dysphonia). H-ABC falls within this spectrum, presenting in the toddler years, typically with dystonia (*Hersheson et al., 2013*), progressive gait impairment, speech and cognitive deficits, as well as characteristic neuroimaging features - hypomyelination and atrophy of the caudate and putamen along with cerebellar atrophy (*van der Knaap et al., 2007*). On human pathological specimens, dorsal striatal areas and the granular layer of the cerebellum exhibit neuronal loss with axonal swelling and diffuse paucity of myelin (*Curiel et al., 2017*; *Simons et al., 2013*). About 65% of published cases with *TUBB4A* mutations have H-ABC; the heterozygous mutation p.

**eLife digest** Inside human and other animal cells, filaments known as microtubules help support the shape of the cell and move proteins to where they need to be. Defects in microtubules may lead to disease. For example, genetic mutations affecting a microtubule component called TUBB4A cause a rare brain disease in humans known as H-ABC.

Individuals with H-ABC display many symptoms including abnormal walking, speech defects, impaired swallowing, and several cognitive defects. Abnormalities in several areas of the brain, including the cerebellum and striatum contribute to these defects. . In these structures, the neurons that carry messages around the brain and their supporting cells, known as oligodendrocytes, die, which causes these parts of the brain to gradually waste away.

At this time, there are no therapies available to treat H-ABC. Furthermore, research into the disease has been hampered by the lack of a suitable "model" in mice or other laboratory animals. To address this issue, Sase, Almad et al. generated mice carrying a mutation in a gene which codes for the mouse equivalent of the human protein TUBB4A.

Experiments showed that the mutant mice had similar physical symptoms to humans with H-ABC, including an abnormal walking gait, poor coordination and involuntary movements such as twitching and reduced reflexes. H-ABC mice had smaller cerebellums than normal mice, which was consistent with the wasting away of the cerebellum observed in individuals with H-ABC. The mice also lost neurons in the striatum and cerebellum, and oligodendrocytes in the brain and spinal cord. Furthermore, the mutant TUBB4A protein affected the behavior and formation of microtubules in H-ABC mice.

The findings of Sase, Almad et al. provide the first mouse model that shares many features of H-ABC disease in humans. This model provides a useful tool to study the disease and develop potential new therapies.

Asp249Asn ($TUBB4A^{D249N/+}$) is the most common mutation (24.1% of overall mutations in a cohort of 166 individuals- personal communication B. Charsar) amongst all forms of $TUBB4A$ associated leu-kodystrophy, and is particularly represented in individuals with a H-ABC phenotype (*Blumkin et al., 2014*; *Ferreira et al., 2014*; *Miyatake et al., 2014*; *Pizzino et al., 2014*; *Purnell et al., 2014*). H-ABC is currently considered an intermediary phenotype, between severely affected early infantile variants and juvenile-adult mild variants (*Nahhas N et al., 2016*).

Although the expression pattern and associated disease phenotypes implicate a functional role of tubulin beta 4A protein in both neurons and oligodendrocytes, little is known about the pathologic mechanisms of $TUBB4A$ mutations. $TUBB4A$ is highly expressed in the central nervous system (CNS), particularly in the cerebellum and white matter tracts of the brain, with more moderate expression in the striatum (*Hersheson et al., 2013*), reflecting disease involvement in H-ABC. At a cellular level, $TUBB4A$ is primarily localized to neurons and oligodendrocytes (OLs), with highest expression in mature myelinating OLs (*Zhang et al., 2014*). Our group has reported the effects of expressing a range of $TUBB4A$ mutations using an OL cell line as well as mouse cerebellar neurons (*Curiel et al., 2017*). Over-expression of the $TUBB4A^{D249N}$ mutation in an OL cell line resulted in decreased myelin gene expression and fewer processes compared to expression of wild type TUBB4A ($TUBB4A^{WT}$) (*Curiel et al., 2017*). Similarly, in cerebellar neurons, $TUBB4A^{D249N}$ over-expression resulted in shorter axons, fewer dendrites, and decreased dendritic branching compared to $TUBB4A^{WT}$ (*Curiel et al., 2017*). Other $TUBB4A$ mutations highlighted phenotypic abnormalities specifically only in neurons and/or OL cell lines, suggesting mutation-specific effects, corresponding to variable clinical phenotypes (*Curiel et al., 2017*). This work highlights the importance of using models with mutations naturally occurring in humans. A spontaneously occurring rat model, the *taiep* rat, with a homozygous p.Ala302Thr *Tubb4a* mutation, has been reported with only a hypomyelinating pheno-type in the brain, optic nerves and certain tracts of the spinal cord but no neuronal pathology (*Duncan et al., 2017*). The *taiep* specific mutation has not been reported in humans but is consistent with our cellular data showing variable cellular phenotypes for different mutations. An interesting feature observed in the *taiep* was accumulation of microtubules, particularly in the OLs, with subsequent demyelination (*Duncan et al., 2017*).

Currently, there are no published animal models for the *TUBB4A*^D249N^ mutation specifically associated with H-ABC; which is key for understanding the pathogenesis and developing therapeutic options for individuals who harbor this mutation. Thus, we have developed a knock-in *Tubb4a*^D249N/D249N^ mouse as a model of H-ABC, recapitulating features of the human disease including dystonia, loss of motor function, and gait abnormalities. The histopathological features of the mouse model include both loss of neurons in striatum and cerebellum and hypomyelination in the brain and spinal cord, as observed in patients (*Curiel et al., 2017*). We have also explored the functional consequence of mutant tubulin on microtubule polymerization and the cell-autonomous role of *Tubb4a* mutation in neurons and OLs of the *Tubb4a*^D249N/D249N^ mice.

## Results

### Generation of a *Tubb4a*^D249N/+^ CRISPR knock-in mice

Heterozygous mutation p.Asp249Asn (*Tubb4a*^D249N/+^) mice were generated using CRISPR-Cas-9 technology by substituting p.Asp249Asn (c.745G > A) mutation in exon 4 of the *Tubb4a* gene. Known off-target effects include one synonymous mutation in cis at p.Lys244Lys (c.732C > A) with the pathogenic variant at p.Asp249Asn (variant at c.745G > A).

*Tubb4a*^D249N/+^ mice were bred to obtain a homozygous *Tubb4a*^D249N/D249N^ mouse colony (*Figure 1A*). Homozygous mice were studied in parallel with *Tubb4a*^D249N/+^ mice, because in a rat model of *Tubb4a* mutation (*Li et al., 2003*), the homozygous animals develop phenotypic manifestations earlier than heterozygous animals. In WT mice, *Tubb4a* gene expression is highest in the cerebellum, spinal cord and striatum (compared to other CNS areas), which are also typically affected brain regions in H-ABC individuals. However, *Tubb4a* gene expression in WT, *Tubb4a*^D249N/+^ and *Tubb4a*^D249N/D249N^ mice are similar in these brain areas (*Figure 1B*), indicating there is no transcriptional change in the face of the mutation.

### *Tubb4a*^D249N/D249N^ mice recapitulate an H-ABC behavioral phenotype

The homozygous *Tubb4a*^D249N/D249N^ mutant mice appear normal compared to WT and heterozygous *Tubb4a*^D249N/+^ littermates until ~post natal (P) day P8. However, by ~P9, *Tubb4a*^D249N/D249N^ mice display tremor (*Figure 1—videos 1* and *2*), which progressively worsens with age (*Figure 1—videos 3* and *4*) until mice become severely ataxic and dystonic (~P21). Additionally, *Tubb4a*^D249N/D249N^ mice show gradual weight reduction from P35-37 compared to *Tubb4a*^D249N/+^ and WT mice (p<0.001, *Figure 1D*).

Motor skills were assessed by measuring ambulation, hanging grip strength, rotarod testing and righting reflexes over time. *Tubb4a*^D249N/D249N^ mice exhibit developmental delay with persistent asymmetric limb movement and crawling gait at P10 relative to ambulation in WT controls (p<0.05, see *Table 1*, *Figure 1E–F*). While *Tubb4a*^D249N/D249N^ mice achieve walking by P14, they display tremor, ataxia and abnormal gait. Ambulatory angle is consistently wider in *Tubb4a*^D249N/D249N^ mice from P14 to P35 (p<0.001, *Figure 1G–H*) suggesting gait instability. Additionally, *Tubb4a*^D249N/D249N^ mice demonstrate early deficits in hanging grip strength test, where compared to WT mice, they fall at a lesser angle (p<0.001, *Figure 1I–J*). Rotarod performance further shows a shorter latency to fall in *Tubb4a*^D249N/D249N^ mice on an accelerating rotarod, with worsening over time (p<0.001, *Figure 1K*, *Figure 1—video 5*). By ~P35-P40, *Tubb4a*^D249N/D249N^ mice are unable to feed themselves, with a decreased righting reflex (p<0.001, *Figure 1L*). At this time, mice reach a compassionate end-point (survival curve, p<0.001, *Figure 1M–N*) and are considered 'end-stage'.

Heterozygous *Tubb4a*^D249N/+^ mice transition well from crawling to ambulation and walk with a normal gait reminiscent of WT mice. Rotarod on *Tubb4a*^D249N/+^ mice at 9 months and 1 year of age demonstrates no motor or gait abnormalities even at later timepoints (*Figure 1—figure supplement 1A*). Further, *Tubb4a*^D249N/+^ mice display similar survival compared to their WT littermates and die because of advanced age (Kaplan-Meier survival curve, *Figure 1—figure supplement 1B*).

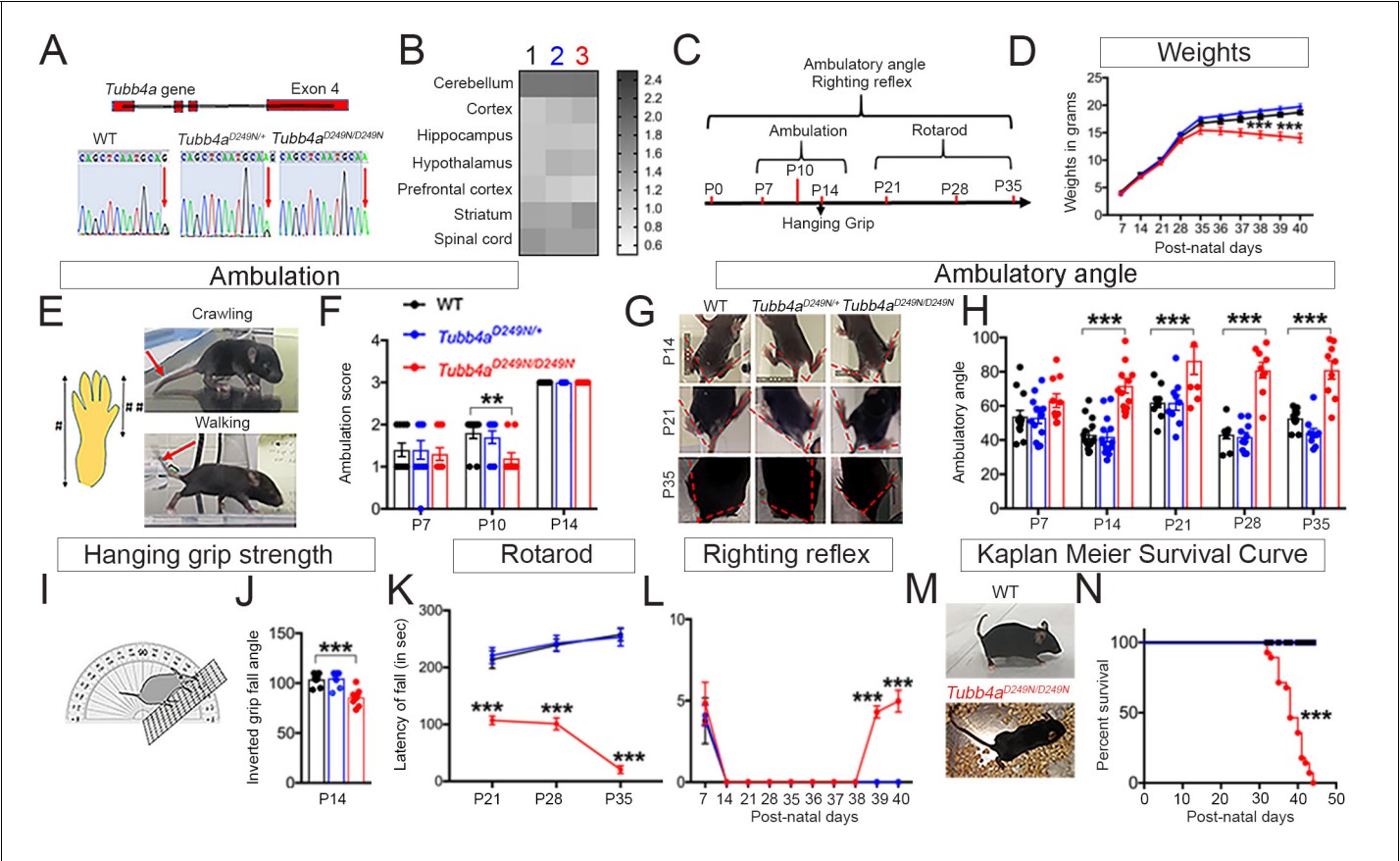

**Figure 1.** *Tubb4a*$^{D249N/D249N}$ mice show decreased survival, gait abnormalities, and progressive motor dysfunction. (**A**) Schematic diagram showing mouse *Tubb4a* gene and sequencing chart of WT, *Tubb4a*$^{D249N/+}$, and *Tubb4a*$^{D249N/D249N}$ mice. Red arrow indicates the position of 745 nucleotide in exon 4; WT shows one peak of 'G', *Tubb4a*$^{D249N/+}$ mice show one peak each of 'G' and 'A', and *Tubb4a*$^{D249N/D249N}$ mice shows two peaks for 'A'. (**B**) Heat map of *Tubb4a* expression in brain;1- WT, 2- *Tubb4a*$^{D249N/+}$ and 3- *Tubb4a*$^{D249N/D249N}$ mice at end-stage (~P35-P40) (n = 3). (**C**) Schematic diagram displaying the time course of behavioral tests. (**D**) Graph for weight measurements of *Tubb4a*$^{D249N/+}$, *Tubb4a*$^{D249N/D249N}$, and WTs from P7, n = 10. (**E**) Illustration of crawling and walking: For ambulation measurement, crawling and walking was scored (see **Table 1**); throughout crawling; the whole hind paw touches the ground as designated by (#), walking; when the toes of the hind paw touch the ground and the heel is elevated, designated by [##]. (**F**) Ambulatory deficits of *Tubb4a*$^{D249N/D249N}$ mice at P7, P10, and P14. Statistical analysis by two-way ANOVA, Tukey post-hoc analysis, n = 10. (**G**) Representative images of ambulatory angles at P14, P21, and P35 of *Tubb4a*$^{D249N/+}$ and *Tubb4a*$^{D249N/D249N}$. (**H**) Ambulatory angle measurements of *Tubb4a*$^{D249N/+}$ and *Tubb4a*$^{D249N/D249N}$ as compared to WT littermates at P7, P14, P21, P28, and P35. Statistical test by two-way ANOVA, post-hoc Tukey test, n = 14. (**I**) Pictorial presentation of hanging grip strength. (**J**) Grip strength measured by inverted fall angle in *Tubb4a*$^{D249N/+}$ and *Tubb4a*$^{D249N/D249N}$ mice. Statistical test by one-way ANOVA, Tukey post-hoc analysis, n = 10. (**K**) Rotarod testing demonstrating latency to fall (in seconds) in *Tubb4a*$^{D249N/+}$ and *Tubb4a*$^{D249N/D249N}$ mice at P21, P28, and P35, n = 14. (**L**) Righting reflex changes of *Tubb4a*$^{D249N/+}$, *Tubb4a*$^{D249N/D249N}$, and WT mice, n = 14. (**M**) Representative image of the end-stage *Tubb4a*$^{D249N/D249N}$ mouse (~P35-P40) with severe dystonia and ataxia relative to WT. (**N**) Kaplan-Meier survival curve of *Tubb4a*$^{D249N/D249N}$ and *Tubb4a*$^{D249N/+}$ mice (Gehan-Breslow-Wilcoxon test, n = 28). Statistical tests performed by repeated measures two-way ANOVA, Tukey post-hoc analysis. Data presented as mean and SEM. *p<0.05, **p<0.01, ***p<0.001. The online version of this article includes the following video, source data, and figure supplement(s) for figure 1:

**Source data 1.** Source files of graphical data of mRNA expression in WT,*Tubb4a*$^{D249N/+}$and*Tubb4a*$^{D249N/D249N}$.

**Source data 2.** Source files of graphical data for Behavioral tests.

**Figure supplement 1.** Normal survival, motor performance, and late-onset hypomyelination in *Tubb4a*$^{D249N/+}$ mice.

**Figure 1—video 1.** Top view of representative WT mouse pup at P9.

https://elifesciences.org/articles/52986#fig1video1

**Figure 1—video 2.** Top view of representative Tubb4aD249N/D249N mouse pup at P9 to demonstrate the onset of tremors.

https://elifesciences.org/articles/52986#fig1video2

**Figure 1—video 3.** Progressive increase in tremors in representative Tubb4aD249N/D249N mouse at P21.

https://elifesciences.org/articles/52986#fig1video3

**Figure 1—video 4.** End-stage to Tubb4aD249N/D249N mouse with severe dystonia and ataxia.

https://elifesciences.org/articles/52986#fig1video4

*Figure 1 continued on next page*

**Figure 1—video 5.** The severe motor function as depicted by rotarod performance of Tubb4aD249N/D249N mouse compared to WT at P35.
https://elifesciences.org/articles/52986#fig1video5

### *Tubb4a$^{D249N/D249N}$* mice exhibit developmental delay in myelination and both *Tubb4a$^{D249N/+}$* and *Tubb4a$^{D249N/D249N}$* mice ultimately show demyelination

Myelination in mice starts at birth in the brainstem and spinal cord and has a nearly adult pattern by P21 (*Baumann and Pham-Dinh, 2001*). Profound myelination deficits are observed in *Tubb4a$^{D249N/D249N}$* mice, relative to their WT littermates, in both corpus callosum and cerebellum at P14 and P21 (Eri-C staining in *Figure 2D and F* and *Figure 2—figure supplement 1* and *Figure 2—figure supplement 2*) with further loss of myelination by end-stage (~P35-P40) (p<0.001, *Figure 2C–F*). Based on Eri-C staining, myelination also decreases in *Tubb4a$^{D249N/+}$* mice at 1 year of age relative to WT littermates (p<0.001, *Figure 2G–H*).

The myelin proteins, PLP and MBP, measured by immunostaining, are also decreased in *Tubb4a$^{D249N/D249N}$* mice relative to *Tubb4a$^{D249N/+}$* and WT mice (*Figure 2W* and *Figure 2—figure supplement 2*, p<0.001), with a significant difference at P21 and end-stage (*Figure 2L, P, T and X*). This decrease in PLP and MBP protein is further confirmed with immunoblotting in forebrain (*Figure 2M–N and U–V*) and cerebellum (*Figure 2Q–R and Y–Z*) of *Tubb4a$^{D249N/D249N}$* mice at P21 (p<0.05) and end-stage (p<0.001). *Tubb4a$^{D249N/+}$* mice have comparable MBP and PLP immunoreactivity as WT mice at P14, P21 (*Figure 2—figure supplements 1* and *2*), and ~P35-P40 (*Figure 2*). However, by 1 year, their myelin protein levels significantly diminish in the corpus callosum compared to WT mice (p<0.05, PLP: *Figure 1—figure supplement 1C–D*; p<0.05, *Figure 2I–J*).

### *Tubb4a$^{D249N/D249N}$* mice show apoptotic oligodendrocyte precursor cells (OPCs) and a dramatic decrease in oligodendrocyte numbers

To understand the developmental and degenerative abnormalities of myelin formation in *Tubb4a$^{D249N/+}$* and *Tubb4a$^{D249N/D249N}$* mice, oligodendrocytes (OLs) were assessed using ASPA stain in the corpus callosum at P14, P21 and end-stage (*Figure 3C*). A dramatic decline occurs in the number of OLs in the *Tubb4a$^{D249N/D249N}$* mice relative to their WT and *Tubb4a$^{D249N/+}$* littermates at all time-points (p<0.001, *Figure 3D*, *Figure 3—figure supplement 1A–B*). However, upon examining the cells double-labeled with CC1 (mature OL marker) and cleaved caspase 3 (cell death marker); no cell death is detected at end-stage (*Figure 3—figure supplement 2A*), potentially ruling out apoptosis mediated OL death. To test, if OL numbers are in fact altered due to an effect on the progenitor pool, the total number of NG2 cells as well as NG2+ Cleaved Caspase 3+ cells were studied for cell death. While the total number of NG2 cells are unchanged (*Figure 3E–F*), a significant number of NG2 cells undergo apoptosis in *Tubb4a$^{D249N/D249N}$* mice compared to WT and *Tubb4a$^{D249N/+}$* mice at P14, P21 and end-stage (p<0.01, p<0.001, *Figure 3G–H* and *Figure 3—figure supplement 2B–C*).

Interestingly, in spite of the ongoing loss, the total number of NG2 and Olig2 populations are stable (*Figure 3—figure supplement 1C–E*). Loss or damage to OPCs can result in a proliferative response (*Hughes et al., 2013*), hence the number of dividing NG2 cells was next assessed at P14,

**Table 1.** Ambulation scores.

Mice were scored using a single trial on crawling, gait symmetry, and limb-paw movement during a straight walk. Ambulation scores were given as mentioned in table.

| Ambulatory skills | Score |
| --- | --- |
| No response | 0 |
| Asymmetric crawling | 1 |
| Symmetric crawling | 2 |
| Walking | 3 |

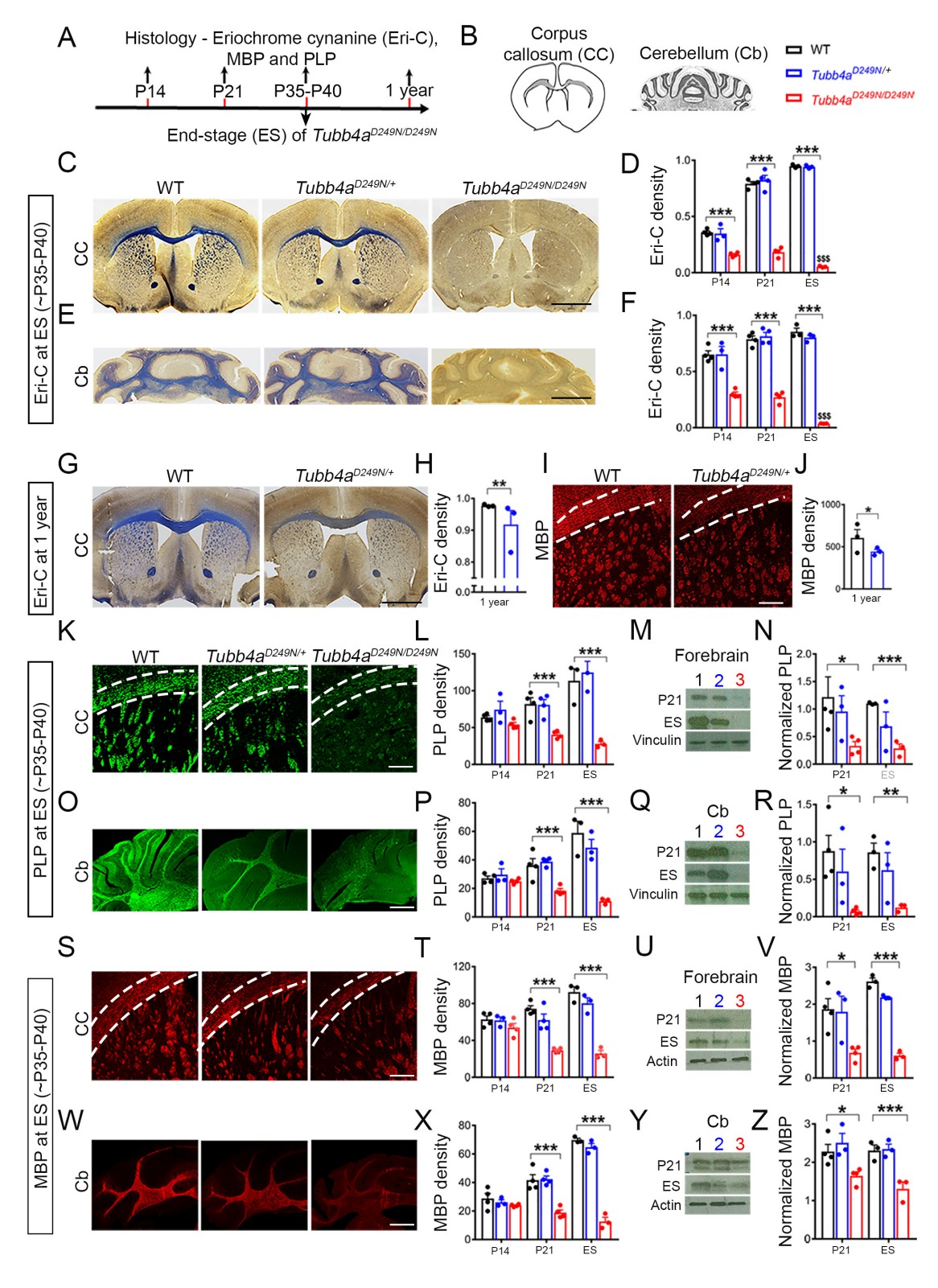

**Figure 2.** *Tubb4a^{D249N/D249N}* mice have severely delayed myelination. (**A–B**) Immunohistochemical assays were performed according to a time course (**A**) in the Corpus callosum (CC) and Cerebellum (Cb) with the analyzed area in grey. (**B**) For each assay, the CC and Cb were imaged and quantified in WT, *Tubb4a^{D249N/+}*, and *Tubb4a^{D249N/D249N}* mice at P14, P21, and at end-stage (ES;~P35-P40), except for Eriochrome cyanine (Eri-C), PLP and MBP which was also assessed at one year in WT and *Tubb4a^{D249N/+}* mice only in CC. (**C–D**) The CC shows significantly decreased Eriochrome cyanine (Eri-C)

*Figure 2 continued on next page*

*Figure 2 continued*

staining (blue) in *Tubb4a*$^{D249N/D249N}$ mice, which worsens over time. (**E–F**) *Tubb4a*$^{D249N/D249N}$ mice show significant and worsening decreased of Eri-C staining in the Cb. (**G–J**) Loss of Eri-C staining (**G–H**) as well as MBP immunostaining (red) (**I–J**) in CC seen in 1-year-old *Tubb4a*$^{D249N/+}$ mice. (**K–N**) PLP immunostaining (green) of CC (**K–L**) and decreased protein levels by western blot in the forebrain from P21 and ES mice (**M–N**) demonstrate a defect in the normal expression of myelin proteins. (**O–R**) A similar effect is seen in PLP immunostaining (**O–P**) and western blot (**Q–R**) in Cb. (**S–Z**) MBP protein staining in CC (**S–T**) and western blot (**U–V**) in the forebrain, as well as the Cb (**W–Z**) also showed a significant and progressive decrease in *Tubb4a*$^{D249N/D249N}$ mice. Statistical tests performed by two-way ANOVA, Tukey post-hoc test or Unpaired t-test. n = 4 mice/group for P14 (except n = 3 for *Tubb4a*$^{D249N/+}$ mice) and P21, n = 3 mice/group for ES and 1 year. Data presented as mean and SEM. *p<0.05 and ***p<0.001. Scale bar = 1 mm (**C, E, G**) or 250 µm (**I, K, O, S, W**). Western blots, lanes; 1, 2, and 3 represent WT, *Tubb4a*$^{D249N/+}$, and *Tubb4a*$^{D249N/D249N}$ mice, respectively.

The online version of this article includes the following source data and figure supplement(s) for figure 2:

**Source data 1.** Source files of graphical data for Myelin quantification.
**Source data 2.** Source files of graphical data for Western blots quantification of MBP and PLP.
**Figure supplement 1.** *Tubb4a*$^{D249N/D249N}$ mice show hypomyelination at P14.
**Figure supplement 2.** *Tubb4a*$^{D249N/D249N}$ mice show hypomyelination at P21.

P21 and end-stage timepoints. A significant increase is detected in NG2+ cells co-labeled with Ki-67 (proliferation marker) in *Tubb4a*$^{D249N/D249N}$ mice at all timepoints compared to *Tubb4a*$^{D249N/+}$ and WT mice (p<0.05, p<0.01; *Figure 3I–J*, *Figure 3—figure supplement 2D–E*), possibly contributing to the steady number of NG2+ and Olig2+ cells. In summary, OPCs undergo cell death and are being replaced in the *Tubb4a*$^{D249N/D249N}$ mice, though possible defects in OPC differentiation results in fewer mature OLs.

## Ultrastructural analysis corroborates myelination deficits in *Tubb4a*$^{D249N/+}$ and *Tubb4a*$^{D249N/D249N}$ mice

Loss of myelination was examined in epoxy sections of optic nerves from *Tubb4a*$^{D249N/D249N}$ mice compared to WT and *Tubb4a*$^{D249N/+}$ mice, at P21 (*Figure 4—figure supplement 1*) and end-stage (data not shown). Electron microscopy further confirms that end-stage *Tubb4a*$^{D249N/D249N}$ optic nerves (n = 3) contain mostly unmyelinated axons and/or axons with myelin sheaths that are inappropriately thin for their axonal caliber (*Figure 4A–C and A'-C'*). In *Tubb4a*$^{D249N/+}$ spinal cords, myelin sheaths are also noticeably thinner than in WT littermates as seen in the epoxy (*Figure 4K–O*) and EM sections (*Figure 4—figure supplement 2*).

Along with dramatically reduced numbers of normally myelinated axons, *Tubb4a*$^{D249N/D249N}$ optic nerves exhibit microglia and/or macrophages containing myelin debris and lipid droplets (*Figure 4D–D'*), dying OLs (*Figure 4E*) as well as vacuoles (*Figure 4G*). As seen in *taiep* rats, we find enlarged astrocytic processes ('astrogliosis'), axons with abnormal accumulations of lysosomes, degenerating unmyelinated and/or myelinated axons, but not bundles of microtubules in OLs (*Duncan et al., 2017*; *Duncan et al., 1992*; *O'Connor et al., 2000*; *Roncagliolo et al., 2006*; *Wilkins et al., 2010*). Many OLs are partially surrounded by unmyelinated axons, some of which are variably surrounded by cytoplasmic invaginations, reminiscent of non-myelinating Schwann cells in peripheral nerves (*Figure 4F*). *Tubb4a*$^{D249N/+}$ optic nerves, in contrast, look similar to WT optic nerves except for more conspicuous astrocytic processes (*Figure 4—figure supplement 1*). These findings are even more striking in *Tubb4a*$^{D249N/D249N}$ spinal cord at end-stage, where most large axons of the ventral funiculus are unmyelinated with conspicuous vacuoles, along with microglia and/or macrophages comprising of myelin debris, lipid droplets, and other unidentified material (*Figure 4—figure supplement 2*).

Compared to WT mice, the g-ratio, a measure of myelin thickness, is significantly higher in optic nerves of *Tubb4a*$^{D249N/D249N}$ and *Tubb4a*$^{D249N/+}$ mice (*Figure 4H*, p<0.001). The axon caliber in optic nerves is significantly decreased in *Tubb4a*$^{D249N/D249N}$ mice versus WT mice at end-stage (*Figure 4I*, p<0.05). Further, plotting g-ratio as a function of axon diameter (*Figure 4J*) demonstrates that intact axons of all calibers in optic nerves have relatively thinner myelin sheaths in *Tubb4a*$^{D249N/+}$ and Tubb4a$^{D249N/D249N}$ mice. The analysis of the percentage of myelination in the optic nerve reveals that ~65% of the axons are unmyelinated (*Figure 4K*, p<0.001) and ~30% of axons are thinly dysmyelinated axons in the *Tubb4a*$^{D249N/D249N}$ (*Figure 4K*, p<0.01) compared to ~3% unmyelinated and 13% thinly dysmyelinated axons in WT mice. While about ~56% of axons are normally myelinated in the *Tubb4a*$^{D249N/+}$ mice (*Figure 4K*, p<0.001),~5% of axons are unmyelinated and

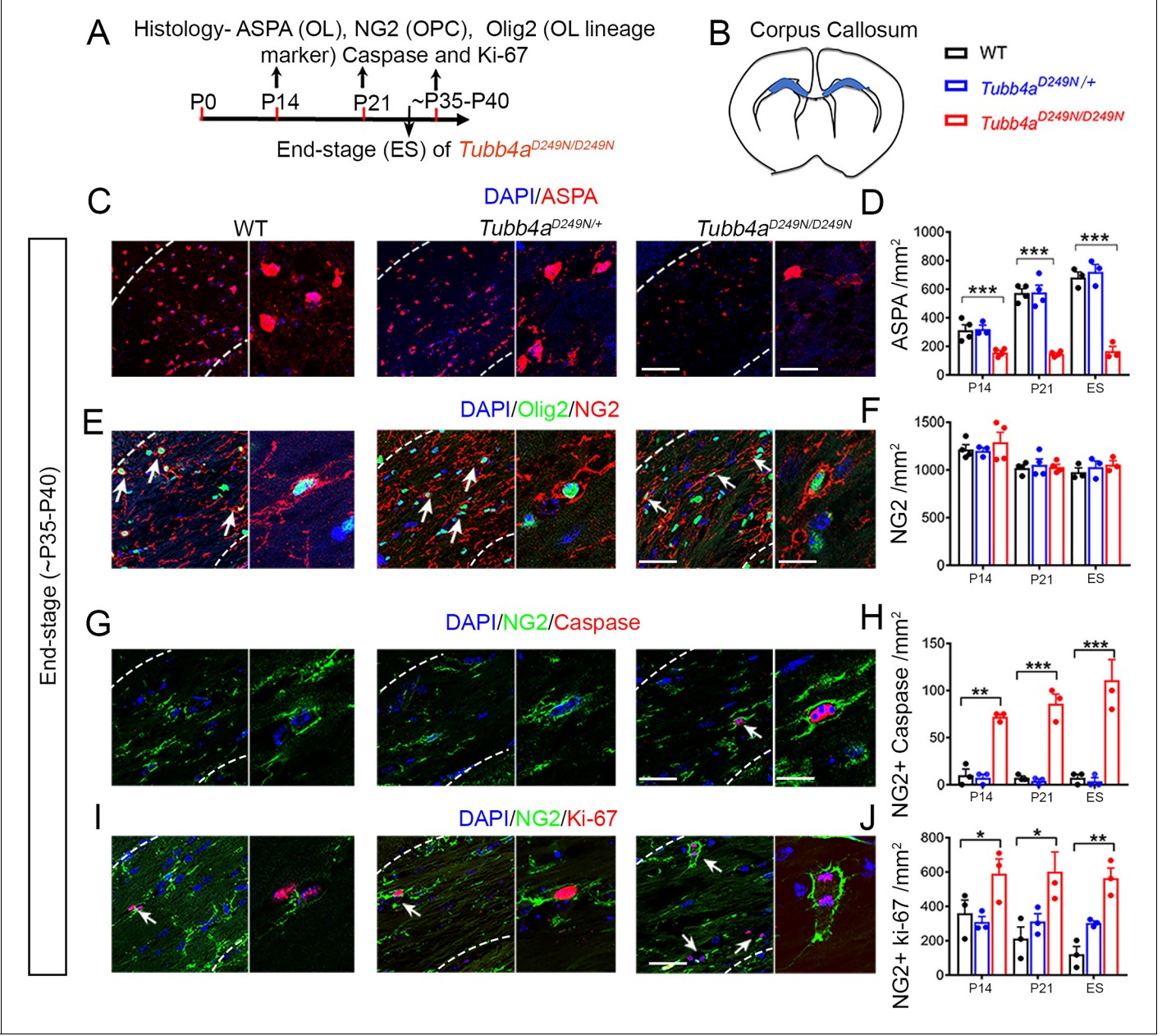

**Figure 3.** *Tubb4a^{D249N/D249N}* mice display reduced number of oligodendrocytes (OLs) and apoptotic Oligodendrocytes progenitor cells (OPC). (**A**) Schematic diagram displaying the time course of immunohistochemical assay. (**B**) Schematic diagram showing area of corpus callosum used to perform counts (blue region). (**C**) Representative images of ASPA-positive OLs in WT, *Tubb4a^{D249N/+}*, and *Tubb4a^{D249N/D249N}* mice at the end-stage (ES;~P35-P40). (**D**) Quantification of ASPA-positive cells/mm² at P14, P21, and ~P35-P40 in WT, *Tubb4a^{D249N/+}*, and ES *Tubb4a^{D249N/D249N}* mice. (**E**) Representative images of double-positive NG2+ Olig2+ cells in WT, *Tubb4a^{D249N/+}*, and *Tubb4a^{D249N/D249N}* mice at ES. (**F**) Quantification of double-positive NG2+ Olig2+ cells/mm² at P14, P21, and ~P35-P40 in WT, *Tubb4a^{D249N/+}*, and ES *Tubb4a^{D249N/D249N}* mice. (**G**) Representative images of double-positive NG2+ caspase cells in WT, *Tubb4a^{D249N/+}*, and *Tubb4a^{D249N/D249N}* mice at ES. (**H**) Quantification of double-positive NG2+ caspase cells/mm² at P14, P21, and ~P35-P40 of WT, *Tubb4a^{D249N/+}*, and ES *Tubb4a^{D249N/D249N}* mice. (**I**) Representative images of double-positive NG2+ Ki-67 cells in WT, *Tubb4a^{D249N/+}*, and *Tubb4a^{D249N/D249N}* mice at ES. (**J**) Quantification of double-positive NG2+ Ki-67 cells/mm² at P14, P21, and ~P35-P40 of WT, *Tubb4a^{D249N/+}*, and ES *Tubb4a^{D249N/D249N}* mice. Statistical test is performed by two-way ANOVA, followed by Tukey post-hoc test. Representative data of two independent experiments with n = 3–4 mice/group for P14 and P21 and n = 3 mice/group for ES. Data is presented as mean and SEM. *p<0.05 and ***p<0.001. Scale bars = 50 µm and 25 µm.

The online version of this article includes the following source data and figure supplement(s) for figure 3:

**Source data 1.** Source files of graphical data for Oligodendrocyte lineage cells in the Corpus callosum.

**Figure supplement 1.** *Tubb4a^{D249N/D249N}* mice show reduced number of oligodendrocytes at P14 and P21.

*Figure 3 continued on next page*

*Figure 3 continued*

**Figure supplement 2.** *Tubb4a^{D249N/D249N}* mice show Oligodendrocyte progenitor cell (OPC) death and proliferation at P14 and P21.

about ~39% thinly myelinated axons (*Figure 4K*, p<0.001). Similar to optic nerves, compared to WT mice, the g-ratio in spinal cords of *Tubb4a^{D249N/D249N}* and in *Tubb4a^{D249N/+}* mice are also significantly different (*Figure 4L–N and O*, p<0.001). Further, plotting g-ratio as a function of axon diameter demonstrates that all axons in spinal cord have relatively thinner myelin sheaths in *Tubb4a^{D249N/+}* and Tubb4a^{D249N/D249N} mice (*Figure 4P*).

## *Tubb4a^{D249N/D249N}* mice demonstrate progressive neuronal loss in cerebellum and striatum

The survival of neurons was examined in the cerebellum and striatum of WT, *Tubb4a^{D249N/+}* and *Tubb4a^{D249N/D249N}* mice with NeuN and Nissl's stain across developmental time points (*Figure 5A*). Nissl's stain on cerebellar sections of *Tubb4a^{D249N/D249N}* mice reveals a severe progressive loss of the granular neuron layer from P21 to end-stage and a remarkable decrease in cerebellar volume (*Figure 5B–C*). Tubb4a^{D249N/D249N} mice display comparable number of granular neurons at P14 compared to WT mice (*Figure 5E* and *Figure 5—figure supplement 1A*), however, a dramatic and progressive granular neuron loss is observed at P21 and end-stage (p<0.001, *Figure 5D–E*). In addition, a significant increase in the number of Cleaved Caspase 3 positive cells co-localized with NeuN is observed at P21 and end-stage, indicating cellular apoptosis (p<0.001, *Figure 5F–G*). The loss of other neuronal populations in the cerebellum, such as Purkinje neurons stained with calbindin, do not display any change in the different groups of mice (*Figure 5—figure supplement 1D–E*).

In the striatum, NeuN counts for *Tubb4a^{D249N/+}* and *Tubb4a^{D249N/D249N}* are comparable to WT mice at P14 and P21 time points (*Figure 5J* and *Figure 5—figure supplement 1B*). But at end-stage, *Tubb4a^{D249N/D249N}* mice demonstrate a significant loss of striatal neurons compared to WT mice (p<0.01, *Figure 5I–J*).

## *Tubb4a* mutation affects oligodendrocyte and neuronal survival in a cell-autonomous manner

In order to explore the effect of *Tubb4a* mutations in OLs and neurons, independent of their in-situ environment, we studied in vitro cultures derived from WT, *Tubb4a^{D249N/+}*, and *Tubb4a^{D249N/D249N}* mice. OPCs were examined using A2B5 stain and a significant decrease is observed in the number of A2B5+ cells isolated from *Tubb4a^{D249N/+}* and *Tubb4a^{D249N/D249N}* mice compared to control WT mice (*Figure 6—figure supplement 1A–D*; p<0.01). Next, to assess if the decrease in OPCs is due to proliferation or cell death, A2B5 cells were co-labeled with Ki-67 or Caspase3 respectively. A significant decrease was noted in the number of A2B5+ Ki-67+ proliferating OPCs in *Tubb4a^{D249N}* (p<0.05) and *Tubb4a^{D249N/D249N}* (p<0.01) group compared to the WT mice (*Figure 6—figure supplement 1A–C*, *Figure 6—figure supplement 1E*). No changes were observed in the number of A2B5+ Caspase3+ cells between the three groups (*Figure 6—figure supplement 1F*). The OPCs were then differentiated towards an OL fate and were examined for PLP, a mature OL marker co-localized with Olig2, a pan OL lineage marker. A significant decrease in the number of PLP+ OLs is observed (*Figure 6A–C*) in *Tubb4a^{D249N/+}* mice (p<0.05) and *Tubb4a^{D249N/D249N}* mice (p<0.01) compared to mature OLs derived from WT mice (*Figure 6E*). However, the total number of Olig2+ cells are similar in all the groups (*Figure 6D*), resulting in a significant reduction in the proportion of mature OLs from the total cells committed to the OL lineage (PLP+/Olig2 cells, *Figure 6F*) in *Tubb4a^{D249N/+}* (p<0.01) and *Tubb4a^{D249N/D249N}* mice (p<0.01) compared to WT mice. These results overall reflect similar changes in OL development as seen in vivo in the mouse tissues and support a cell-autonomous contribution of *Tubb4a^{D249N/+}* mutation in the development of OL lineage cells.

We also examined cortical neurons using in vitro cultures derived from WT, *Tubb4a^{D249N/+}* and *Tubb4a^{D249N/D249N}* mice. We analyzed survival based on the number of neurons labeled with Tuj1 and MAP2 stains at one-week post-plating. *Tubb4a^{D249N/D249N}* neurons display a significant decrease in survival relative to WT neurons (*Figure 6G–I*, p<0.01) and *Tubb4a^{D249N/+}* neurons. We also assessed whether *Tubb4a^{D249N}* affects neuronal morphology, by measuring axonal outgrowth and total dendritic length. Axon length is shorter in *Tubb4a^{D249N/+}* neurons (*Figure 6J*) and

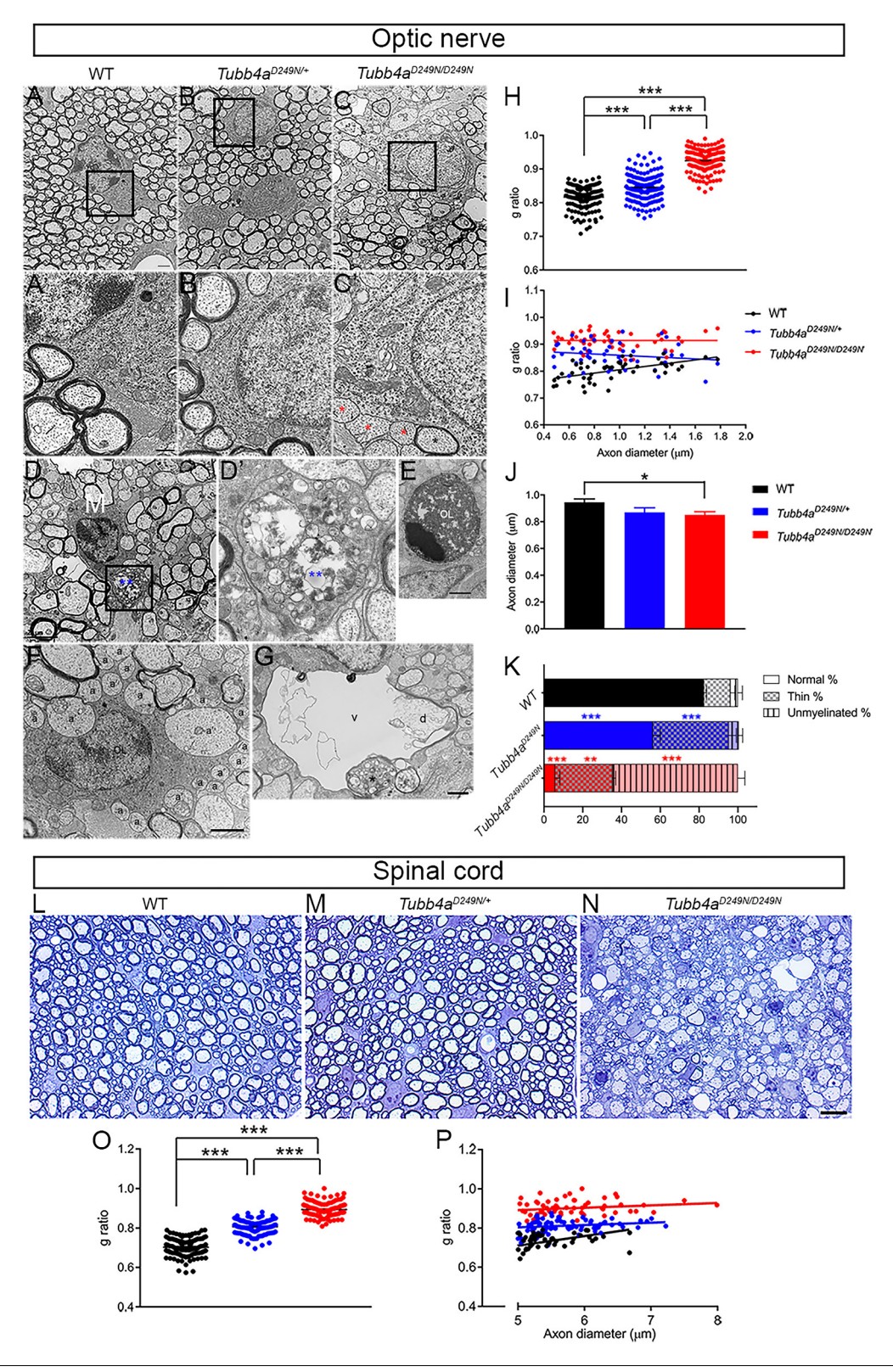

**Figure 4.** Ultrastructural myelin abnormalities in optic nerves and spinal cord of *Tubb4a^D249N/+* and *Tubb4a^D249N/D249N* mice. (A–C) Representative electron microscopic (EM) images of optic nerves from WT, *Tubb4a^D249N/+*, and *Tubb4a^D249N/D249N* mice at end-stage (~P35) as indicated. Scale bar = 800 nm. The regions bounded by the rectangles are shown at higher magnification in (A'–C'), and illustrate the myelin sheath thickness. Red asterisks = unmyelinated axons, black asterisk = thinly myelinated axon. Scale bar = 400 nm. (D) Microglial cell/macrophage (M) containing

*Figure 4 continued on next page*

*Figure 4 continued*

phagocytosed material (blue asterisks) in *Tubb4a^D249N/D249N^*. The region bounded by rectangle is shown at higher magnification in D'. Scale bar = 2 µm. (**E**) This electron microscopic image shows an apoptic nucleus, presumably an oligodendrocyte (OL). Scale bar = 1 µm. (**F**) This EM image shows unmyelinated axons that appose (a) or are surrounded by (a') processes of an oligodendrocyte (OL). (**G**) A degenerating axon (d) with an incomplete axolemma, in a vacuole (v). The asterisk marks an axon containing vesicles. (**H**) g-ratios in optic axons from WT, *Tubb4a^D249N/+^*, and *Tubb4a^D249N/D249N^* mice. (**H**) g-ratio plotted against axon diameter for WT, *Tubb4a^D249N/+^*, and *Tubb4a^D249N/D249N^* mice. (**I**) Optic axon diameters (mean and SEM) for WT, *Tubb4a^D249N/+^*, and *Tubb4a^D249N/D249N^* mice; 50 axons per animals. (**K**) % myelination in the optic nerve of end stage mice (**L–N**) Representative semi-thin sections of spinal cord from WT, *Tubb4a^D249N/+^*, and *Tubb4a^D249N/D249N^* mice at end-stage (~P35) as indicated. Scale bar = 10 µm. (**O**) g-ratios in ventral funiculus of spinal cord from WT, *Tubb4a^D249N/+^*, and *Tubb4a^D249N/D249N^* mice. (**P**) g-ratio plotted against axon diameter in ventral funiculus of spinal cord from WT, *Tubb4a^D249N/+^*, and *Tubb4a^D249N/D249N^* mice. (n = 3 animals per group). One-way ANOVA was performed on the data set followed by Tukey post-hoc test. Data is presented as mean and SEM. *p<0.05, **p<0.001, ***p<0.001.

The online version of this article includes the following source data and figure supplement(s) for figure 4:

**Source data 1.** Source files of graphical data for Electron microscopy.
**Figure supplement 1.** Hypomyelination and axon loss in *Tubb4a^D249N/D249N^* optic nerve at P21.
**Figure supplement 2.** Hypomyelination in *Tubb4a^D249N/+^* and *Tubb4a^D249N/D249N^* spinal cords at end-stage (~P35-P40).

significantly shorter in *Tubb4a^D249N/D249N^* neurons than WT neurite length (p<0.05). Total dendritic length is not significantly altered across genotypes (*Figure 6K*), although *Tubb4a^D249N/D249N^* neurons shows a trend for shorter dendritic arbors. Overall, H-ABC associated *Tubba^D249N^* mutation affects neuronal development, particularly axonal elongation.

## *Tubb4a^D249N/+^* and *Tubb4a^D249N/D249N^* mutation alters microtubule dynamics in axons

We next investigated whether *Tubb4a* mutation affects microtubules dynamics in the distal axons of cortical neurons expressing mCherry-tagged EB3, a protein that forms comet-like structures at the growing plus-ends of microtubules from WT, *Tubb4a^D249N/+^* and *Tubb4a^D249N/D249N^* mice (*Figure 7A*; *Stepanova et al., 2003*). The average number of EB3 comets is not significantly different in neurons from WT, *Tubb4a^D249N/+^*, or *Tubb4a^D249N/D249N^* mice (*Figure 7B*). However, we noted that the *Tubb4a^D249N/D249N^* data shows a greater variance (Browne-Forsythe test for equality of variances revealed a significant difference among the conditions, p<0.05) compared to WT and *Tubb4a^D249N/+^* neurons. Histogram analysis reveals two distinct populations within the *Tubb4a^D249N/D249N^* dataset, with some neurons displaying a lower density of EB3-comets and others a higher density of comets than seen in neurons from either the WT or heterozygous mice (*Figure 7C*). EB3 comet run-time (*Figure 7D*) in *Tubb4a^D249N/D249N^* neurons is significantly shorter (p<0.001) compared to WT neurons and *Tubb4a^D249N/+^* neurons. Similarly, EB3 comet run-length in *Tubb4a^D249N/D249N^* and *Tubb4a^D249N/+^* neurons is significantly shorter than in WT neurons (p<0.001, *Figure 7E*). In contrast, EB3 comet velocity is similar across all genotypes (*Figure 7—figure supplement 1*), indicating that microtubule growth rates were not affected.

## Discussion

*TUBB4A* mutations were identified in H-ABC affected individuals in 2013 (*Simons et al., 2013*), and additional mutations were later associated with a spectrum of hypomyelinating conditions (*Hamilton et al., 2014*; *Nahhas N et al., 2016*). The p.Asp249Asn (D249N) variant has been closely tied to the classic features of H-ABC, including the characteristic triad of hypomyelination with striatal and cerebellar atrophy. Although the naturally occurring *taeip* rat model harbors a *Tubb4a* mutation, the p.Ala302Thr mutation has not previously been seen in affected human individuals; the model lacks cerebellar and striatal atrophy (*Duncan et al., 2017*; *Li et al., 2003*) and is more consistent with those individuals manifesting isolated hypomyelination (*Pizzino et al., 2014*). In this study, we sought to fully explore H-ABC in a novel CRISPR-Cas9 transgenic model of the classic p.Asp249Asn (D249N) mutation, hoping to provide greater understanding of the diverse cellular phenotypes affected in the human disease.

*Tubb4a* mutations in our model results in a severe myelin pathology, consistent with hypomyelination and progressive demyelination over time. Profound hypomyelination may contribute to early onset tremor in *Tubb4a^D249N/D249N^* mice, similar to what is seen in *Shiverer* (*Biddle et al., 1973*),

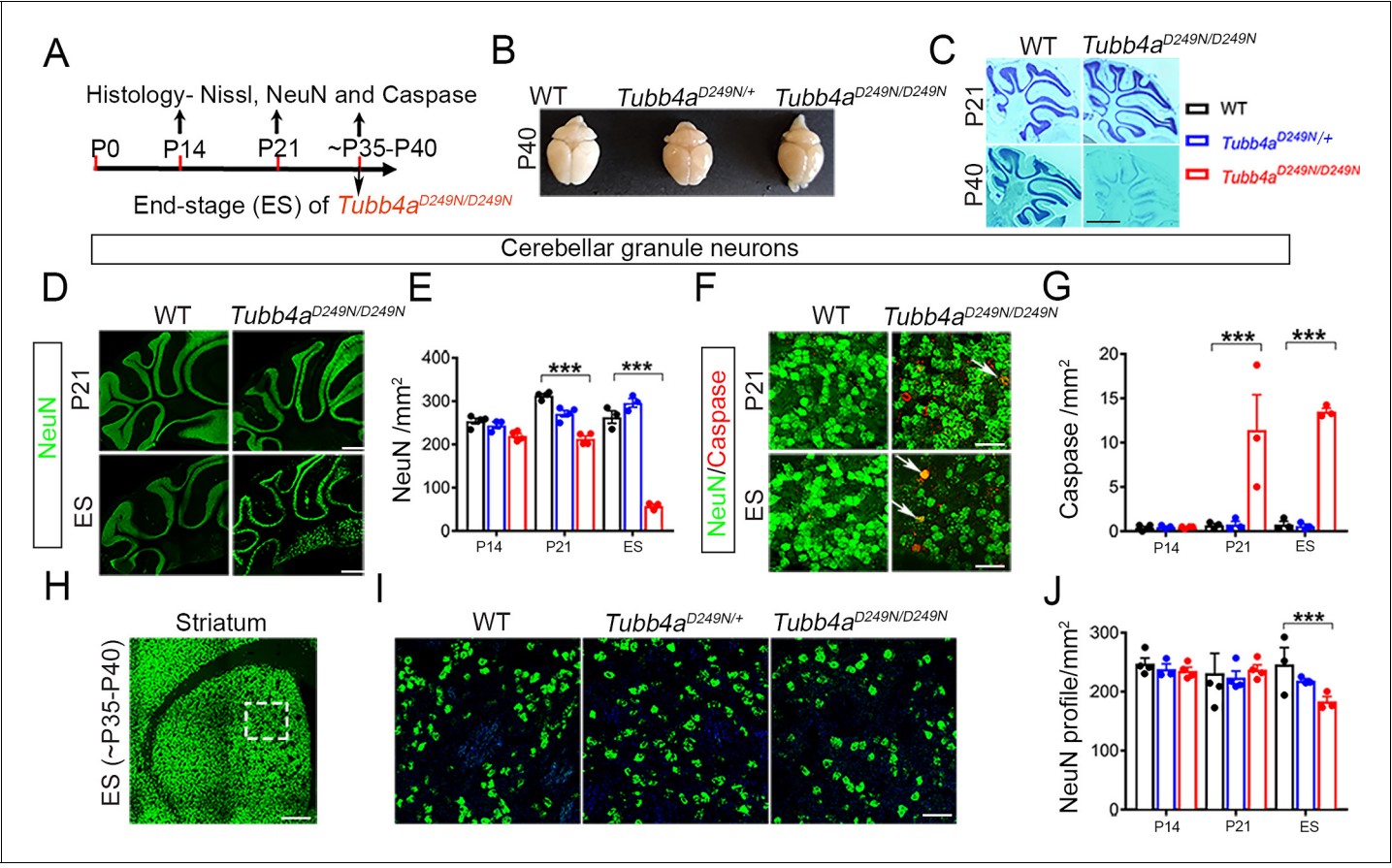

**Figure 5.** *Tubb4a^{D249N/D249N}* mice show severe cerebellar granular neuronal loss and significant striatal neuronal loss. (**A**) Schematic diagram displaying the time course of immunohistochemical assays. (**B**) Schematic diagram showing the whole brain mounts of WT, *Tubb4a^{D249N/+}*, and *Tubb4a^{D249N/D249N}* mice at P40. (**C**) Nissl stain images of cerebellum at P21 and P40 of WT and *Tubb4a^{D249N/D249N}* mice. (**D**) Representative images of NeuN (green) showing cerebellar granular neurons at P21 and End-stage (ES;~P35-P40) of WT and *Tubb4a^{D249N/D249N}* mice. (**E**) Quantification of cerebellar granular neuron (NeuN; green) counts/mm² at P14, P21, and ES. (**F**) Representative images of double immuno-positive cerebellar granule neurons stained by NeuN+ (green) and cleaved caspase 3+ (red) (as shown by white arrow) at P21 and ES of WT and *Tubb4a^{D249N/D249N}* mice. (**G**) Quantification of double-positive NeuN+ (cerebellar granule neurons) and caspase+ (red) counts/mm² at P14, P21, and ES. (**H**) Schematic diagram of striatum showing area used for quantifying neuronal counts (by dashed box). (**I**) Representative images of striatal neurons stained by NeuN (green) at ES of WT, *Tubb4a^{D249N/+}*, and *Tubb4a^{D249N/D249N}* mice. (**J**) Quantification of striatal neuronal counts/mm² at P14, P21, and ES in WT, *Tubb4a^{D249N/+}*, and *Tubb4a^{D249N/D249N}* mice. Statistical test was performed by two-way ANOVA, followed by Tukey post-hoc test. Representative data of two independent experiments with n = 4 mice/group for P14 and P21 (except n = 3 for *Tubb4a^{D249N/+}* for P14 time point) and n = 3 mice/group for ES. Scale bar = 1 mm (C, D, H) or 25 μm (F) or 250 μm (I). Data is presented as mean and SEM. *p<0.05 and ***p<0.001.

The online version of this article includes the following source data and figure supplement(s) for figure 5:

**Source data 1.** Source files of graphical data for Cerebellar granule neurons and striatal neurons quantification.
**Figure supplement 1.** *Tubb4a^{D249N/D249N}* mice show comparable Purkinje neuronal counts.

*Shi^{mild}* (**Readhead and Hood, 1990**) and *Jimpy* mice (**Nave et al., 1986**). *Tubb4a* is highly expressed in mature OLs (**Zhang et al., 2014**), which may contribute to the significant glia phenotype in this mouse. The number of OPCs appears to be preserved in vivo in *Tubb4a^{D249N/D249N}* mice, although co-staining of NG2 (OPC marker) with caspase indicates increased cell death in early OL lineage cells of *Tubb4a^{D249N/D249N}* mice. A preserved overall number of NG2 and Olig2+ cells suggests that the death of OPCs may result in increased proliferation in vivo, supported by co-staining of NG2 with Ki-67. It has previously been shown that OPCs are dynamic cells and maintain their homeostatic behavior since NG2 cells abolished by differentiation or death are quickly restored through proliferation of the nearest OPCs (**Hughes et al., 2013**). OPCs isolated from *Tubb4a^{D249N/+}* and *Tubb4a^{D249N/D249N}* mice showed no changes in cell death markers but displayed defects in proliferation with fewer A2B5+ Ki-67+ cells compared to control WT control mice. Thus in vitro changes reflect a cell-

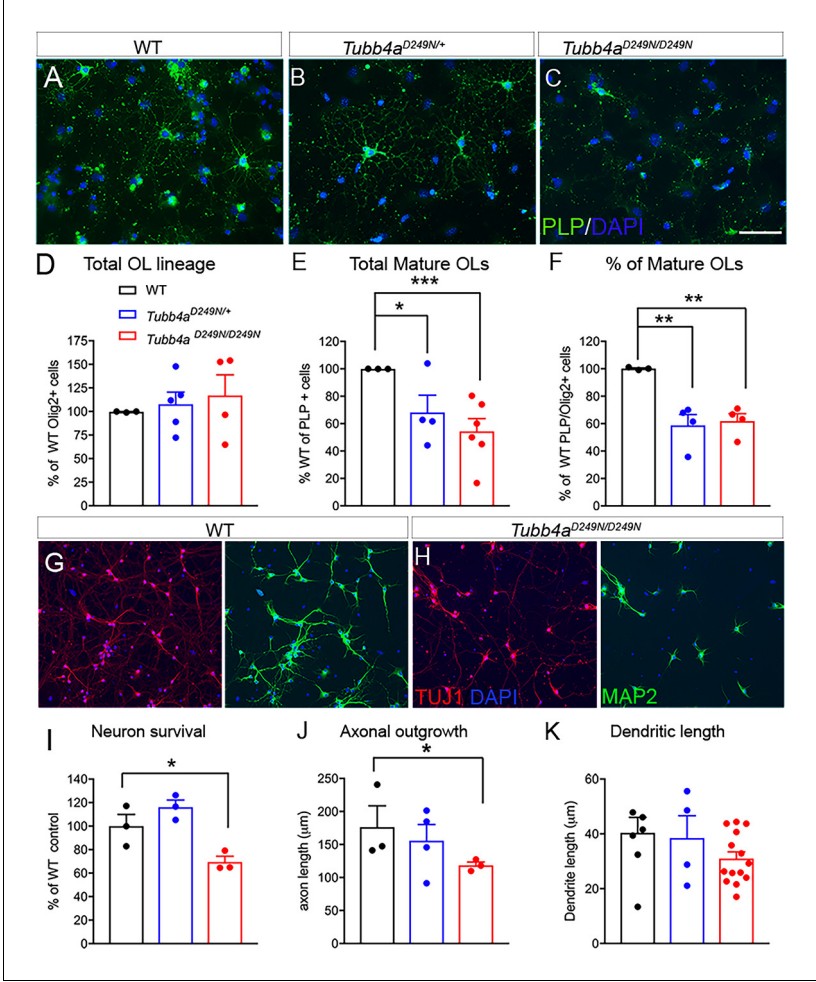

**Figure 6.** Oligodendrocytes and neurons from *Tubb4a*$^{D249N/+}$ and *Tubb4a*$^{D249N/D249N}$ mice display reduced branching and processes. (**A–C**) Representative images of PLP labeled oligodendrocytes (OL) isolated from WT, *Tubb4a*$^{D249N/+}$, and *Tubb4a*$^{D249N/D249N}$ mice. Scale bar = 50 μm. (**D**) The number of Olig2 labeled cells were counted in coverslips from WT, *Tubb4a*$^{D249N/+}$, and *Tubb4a*$^{D249N/D249N}$ mice and plotted as percentage of Olig2+ cells in WT animals. (**E**) Total number of PLP+ cells quantified from WT, *Tubb4a*$^{D249N/+}$, and *Tubb4a*$^{D249N/D249N}$ mice were plotted as a percentage of PLP+ cells in WT animals. (**F**) Number of mature PLP+ cells from the total Olig2+ cells were plotted as a percent of WT animals. The experiments were repeated at least three independent times (n = 3). (**G**) Representative images of cortical neurons from WT mice stained with Tuj1 (axonal marker) and MAP2 (dendritic marker). (**H**) Representative images of cortical neurons from *Tubb4a*$^{D249N/D249N}$ mice stained with Tuj1 and MAP2. Scale bar = 75 μm. (**I**) Number of surviving neurons at 1 week post-plating were quantified and plotted as percent of WT neurons. (**J**) Axon length was measured using Neurite tracer plugin and plotted for all groups. (**K**) Dendritic length was measured using Neurite tracer plugin and plotted for all groups. All experiments were conducted at least three independent times, with n = 1–3 pups/genotype and technical replicate of n = 3/ pup. Data is presented as mean and SEM. One-way ANOVA was performed on the data set followed by Tukey post-hoc test. *p<0.05, ***p<0.001.

The online version of this article includes the following source data and figure supplement(s) for figure 6:

**Source data 1.** Source files of graphical data for Oligodendrocyte and neuron culture.
**Figure supplement 1.** *Tubb4a*$^{D249N/D249N}$ mice show decreased proliferation and no change in OPC death in vitro.

autonomous effect resulting in decreased proliferation with no change in cell death. Conversely, in vivo dynamics of OPCs is complex as the increase in OPC proliferation occurs in the context of cell death, which could possibly be due to cross-talk with other cells such as neurons (*Nagy et al., 2017*) and influence of the milieu, as has been seen in other neurodegenerative disorders (*Fernandez-*

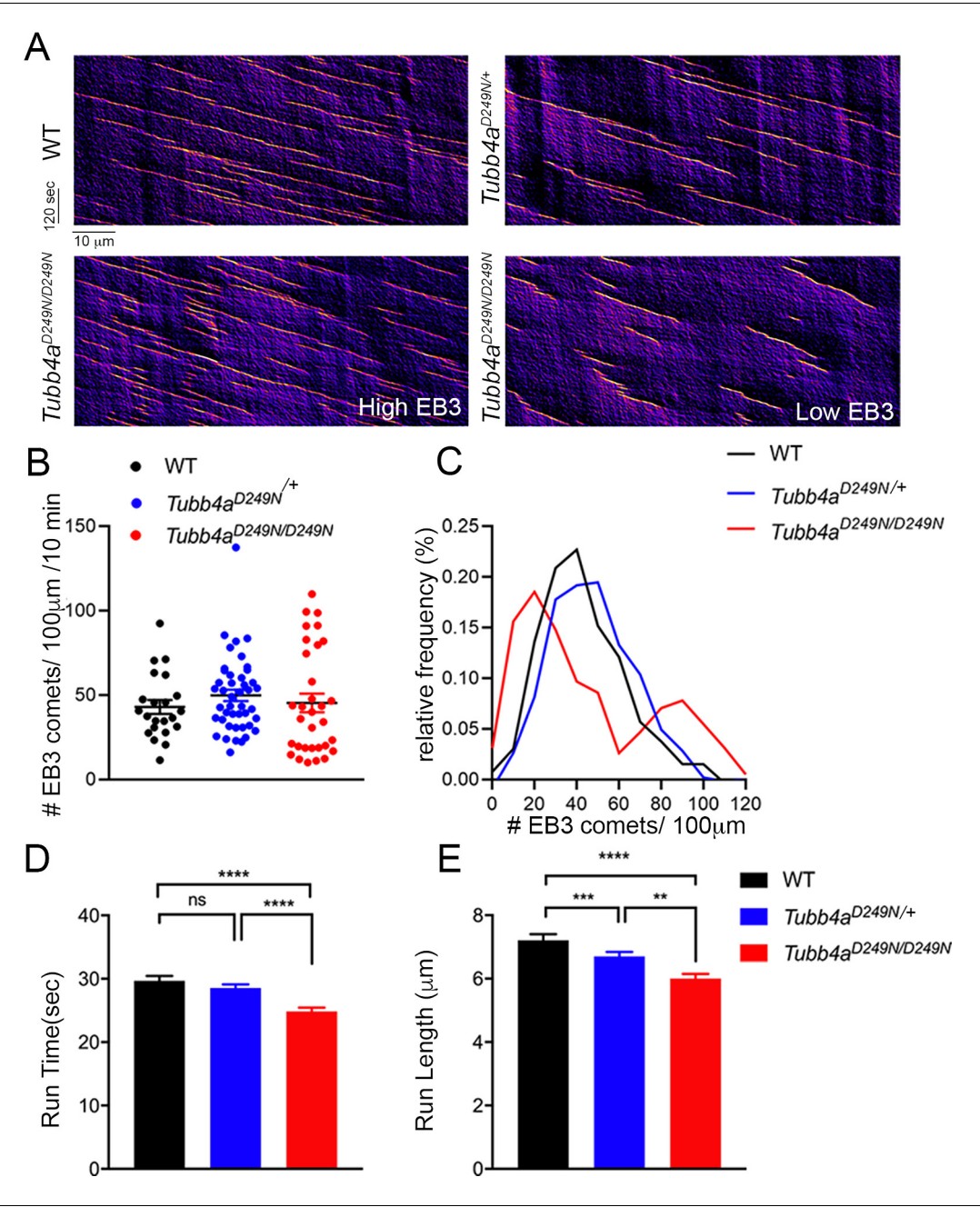

**Figure 7.** Microtubule polymerization is affected in *Tubb4a*$^{D249N/+}$ and *Tubb4a*$^{D249N/D249N}$ mice. (**A**) Example kymographs generated based on EB3 tracking from WT, *Tubb4a*$^{D249N/+}$, and *Tubb4a*$^{D249N/D249N}$ cortical neurons. EB3 comet frequency was not significantly different (**B**) but a histogram depicting EB3 comet run-time relative frequency (**C**) suggests two different populations of EB3 comets. (**D**) Run-length and (**E**) Run-time of EB3 comets plotted for WT, *Tubb4a*$^{D249N/+}$, and *Tubb4a*$^{D249N/D249N}$ were significantly shorter in *Tubb4a*$^{D249N/+}$ and *Tubb4a*$^{D249N/D249N}$ mice. All experiments were conducted at least three independent times, with n = 1–3 pups/ genotype and technical replicate of n = 3/pup. Data is presented as mean and SEM. One-way ANOVA was performed on the data set followed by Tukey post-hoc test. *p<0.05, **p<0.001, ***p<0.001.

The online version of this article includes the following source data and figure supplement(s) for figure 7:

**Source data 1.** Source files of graphical data for Microtubule dynamics data.
**Figure supplement 1.** *Tubb4a*$^{D249N/+}$ and *Tubb4a*$^{D249N/D249N}$ mice show altered MT dynamics.

*Castaneda and Gaultier, 2016*; *Kang et al., 2013*; *Lee et al., 2019*; *McTigue and Tripathi, 2008*; *Tripathi and McTigue, 2007*). Ultimately, both in vitro and in vivo studies show significant decrease in the number of mature OLs in $Tubb4a^{D249N/D249N}$ mice as opposed to WT control mice. Since OLs express a greater degree of Tubb4a, it is possible a proportion of mature OLs die, not captured by the apoptotic marker caspase in vivo or die by a non-apoptotic mechanism. Thus, both cell death and inefficient differentiation may account for severe deficits seen in formation of mature OLs in $Tubb4a^{D249N/D249N}$ mice.

Pathology does not appear to be restricted to the OL lineage in H-ABC, however. $Tubb4a^{D249N/D249N}$ mice show evidence of severe loss of cerebellar granular neurons at P21 and more limited striatal neuron degeneration after ~P37. The milder defects seen in striatal neurons may be related to the variable expression of *Tubb4a*, that is relatively higher in cerebellum than striatum (*Figure 1B*; *Hersheson et al., 2013*). We also studied cortical neurons in culture from $Tubb4a^{D249N/+}$ and $Tubb4a^{D249N/D249N}$ mice, demonstrating decreased survival and stunted axonal and dendritic length, although no obvious cortical defect is identified in vivo. Neuronal loss in the striatum and cerebellar granule layer is consistent with reported neuropathology in individuals affected by H-ABC (*Pizzino et al., 2014*; *van der Knaap et al., 2002*) and may underlie the progressive gait abnormalities, ataxia, and motor dysfunction seen in both this mouse model and affected individuals.

MTs are integral to neuronal and OL development, contributing to their structure, including arborization, polarity, growth cone dynamics; as well as function such as axonal transport and intracellular transport of key myelin proteins through efficient MT polymerization (*Clark et al., 2016*). A number of mutations in α-tubulin and β-tubulin are attributed to neurological disorders such as Parkinson's disease (*Cartelli et al., 2010*) and Amyotrophic Lateral Sclerosis (*Fanara et al., 2007*) due to altered MT dynamics resulting in impaired axonal transport crucial for active cargo transport. Previous in vitro reports (*Curiel et al., 2017*; *Vulinovic et al., 2018*) suggest that $TUBB4A^{D249N/+}$ causes alterations in tubulin dynamics. In current studies, while there is no change in microtubule assembly rates in functional studies in $Tubb4a^{D249N/+}$ and $Tubb4a^{D249N/D249N}$ neurons, a potential increase occurs in the frequency of pausing at the growing microtubule ends. Our data also suggests that there may be alterations in microtubule organization in axons, given the increased variance in EB3-comet density along the axons of $Tubb4a^{D249N/D249N}$ neurons. Alterations in MT dynamics or organization in $Tubb4a^{D249N/D249N}$ neurons may hamper the transport of cargo required for axon elongation and dendritic branching. Additionally, given the complexity of OL processes and myelin sheath development, it can be foreseen that inefficient delivery of cargo along MTs also contributes to the decreased maturation and complexity of $Tubb4a^{D249N/D249N}$ OLs.

Notably, the *taeip* rat displays an accumulation of microtubules in OLs, unmyelinated axons, and astrogliosis along with perinuclear localization of RNA for PLP, MAG and MBP myelin genes, which was attributed to altered activity of the motor protein dynein for MBP trafficking in mutated OLs (*Duncan et al., 2017*; *O'Connor et al., 2000*; *Song et al., 2003*; *Wilkins et al., 2010*). Our EM studies in $Tubb4a^{D249N/D249N}$ mice show unmyelinated or abnormally thin myelinated axons, astrogliosis, axons with abnormal accumulation of lysosomes, microglia, and/or macrophages containing myelin debris, and lipid droplets reminiscent of deficits seen in *taiep* rats. However, the remarkable microtubule accumulation demonstrated in the *taiep* rat (*Duncan et al., 2017*) is not seen in our $Tubb4a^{D249N/D249N}$ mice.

Despite no microtubule accumulation, it is noteworthy that the $Tubb4a^{D249N/D249N}$ mutation appears to significantly affect microtubule dynamics in our model. This may be the basis for the cell-autonomous effects of *Tubb4a* mutations, leading to decreased OL maturation, impaired axonal growth and reduced neuronal survival. The exact mechanism by which $Tubb4a^{D249N/+}$ mutation impacts microtubule function, myelination, and neuronal function remains unknown and needs to be further explored.

An important limitation of this study is the use of mice with both heterozygous and homozygous mutations, when in humans the inheritance pattern of H-ABC is sporadic with heterozygous mutations causing disease. While both $Tubb4a^{D249N/D249N}$ and heterozygous $Tubb4a^{D249N/+}$ mice demonstrate severe hypomyelination followed by additional myelin loss; only the homozygous $Tubb4a^{D249N/D249N}$ mice show early deficits in gait and motor skills, consistent with ataxia and tremor seen in H-ABC affected individuals. One potential explanation for this species-specific difference is dosage sensitivity, resulting in a dissimilar penetrance of phenotypes. This has been reported for the *taiep* rat model (*Li et al., 2003*); in addition, there are several other reported genes such as

GATA3 (*Lim et al., 2000*), TBX1 (*Jerome and Papaioannou, 2001*), GLI3 (*Böse et al., 2002*) where, heterozygous mutations are present in humans but homozygous mice have similar phenotypic expression to the human disease. The presence of disease in both heterozygous and homozygous animals could be due to mutations causing either loss or gain of function. However, our previous in vitro work and a study in human SH-SY5Y cells overexpressing mutant TUBB4A (*Curiel et al., 2017*; *Vulinovic et al., 2018*) suggests a mechanism for toxic gain of function. This is further supported by existing *Tubb4a knock out* (KO) mouse models. *Tubb4a* KO mice with LacZ expression (*Skarnes et al., 2011*) are available at the knock-out mouse project repository (http://www.mouse-phenotype.org/data/genes/MGI:107848#section-associations). Homozygous *Tubb4a* KO mice show normal embryonic development and growth relative to WT mice. Phenotypic lac Z expression data shows that the nervous system appears normal with no discernible cerebellar neuron loss. Together this body of evidence suggests a mechanism of dominant toxic gain of function and while presence of WT Tubb4a might not be essential for brain development and function, mutations may later result in neuronal and oligodendrocyte deficits.

As various *TUBB4A* mutations have been reported (*Lu et al., 2017*; *Pizzino et al., 2014*; *Tonduti et al., 2016*), it is becoming recognized that mutation specific cellular effects, with independent involvement of the striatum, myelinating cells, and cerebellum, may be responsible for the wide phenotypic variability seen in this condition (*Curiel et al., 2017*; *Pizzino et al., 2014*). $Tubb4a^{D249N/D249N}$ mice are the first model to demonstrate both neuronal and oligodendroglial defects, and replicate the behavioral and neurodegenerative features of classical H-ABC disease. These data support a pathogenic pathway in which altered microtubules are critical drivers of disease pathogenesis across diverse cellular populations. $Tubb4a^{D249N/D249N}$ mice provide a key tool to explore the molecular mechanisms of this complex disease and test the efficacy of therapeutic strategies.

## Materials and methods

### Key resources table

| Reagent type (species) or resource | Designation | Source or reference | Identifiers | Additional information |
|---|---|---|---|---|
| Genetic reagent (*Mus musculus*) Background: C57BL/6J | *Tubb4a* Accession number: NM_009451.3 | Mouse model designed at Cyagen | Mouse *Tubb4a* Knockin Project (CRISPR/Cas9) with p.Asn249Asn (D249N) mutation | Target region of mouse Tubb4a locus modified and D249N mutation was introduced ($Tubb4a^{D249N/+}$) |
| Sequence-based reagent | gRNA sequence 1 (Matches forward strand of *Tubb4a* gene) | Cyagen designed | | CAATGCAGATCTACGCAAGCTGG |
| Sequence-based reagent | gRNA sequence 2 (Matches reverse strand of *Tubb4a* gene) | Cyagen designed | | CAATGCAGATCTACGCAAGCTGG |
| Mouse genotyping And Sequence based reagents | To identify the genotype of $Tubb4a^{D249N/+}$ mouse | Taq-Takara | PCR forward and reverse primers and Sequencing | 5'CCGAGAGGAGTTTCC AGACAGACAGGATC3' 5'GCTCTGCACACTT AACATCTGCTCG 3' |
| Antibody | anti- PLP (Rat monoclonal) | IDDRC hybridoma, courtesy Dr. Judith Grinspan | RRID:AB_2827948 | Dilution Used for IF: 1:1 Dilution Used for Western blot: 1:1000 |
| Antibody | anti-MBP (Rabbit polyclonal) | Abcam | Cat#: ab40389 RRID:AB_1141521 | Dilution Used for IF: 1:250 Dilution Used for Western blot: 1:1000 |
| Antibody | anti-NG2 (Rabbit polyclonal) | US biological | Cat#: C5067-70D RRID:AB_2827946 | Dilution Used for IF: 1:250 |
| Antibody | anti-NG2 (Mouse monoclonal) | Thermo Fisher Scientific | Cat#: #37–2700 RRID:AB_2533307 | Dilution Used for IF: 1:100 |

*Continued on next page*

*Continued*

| Reagent type (species) or resource | Designation | Source or reference | Identifiers | Additional information |
|---|---|---|---|---|
| Antibody | anti-Olig2 (Rabbit polyclonal) | Millipore | Cat#: MABN50 RRID:AB_10807410 | Dilution Used for IF: 1:100 |
| Antibody | anti-NeuN (Mouse monoclonal) | Millipore | Cat#: MAB377 RRID:AB_2298772 | Dilution Used for IF: 1:100 |
| Antibody | anti-cleaved Caspase 3 (Rabbit polyclonal) | Cell signaling | Cat#: #9579 RRID:AB_10897512 | Dilution Used for IF: 1:100 |
| Antibody | anti- Ki-67 (Rabbit polyclonal) | Thermo Fisher Scientific | Cat#: #RM9106S0 RRID:AB_2341197 | Dilution Used for IF: 1:100 |
| Antibody | anti-calbindin (Rabbit polyclonal) | Swant | Cat#: CB38 RRID:AB_2721225 | Dilution Used for IF: 1:250 |
| Antibody | anti-A2B5 (Mouse monoclonal) | IDDRC hybridoma, courtesy Dr. Judith Grinspan | RRID:AB_2827951 | Dilution Used for IF: 1:1 |
| Antibody | anti-MAP2 (Mouse monoclonal) | Sigma | Cat#: 1406 RRID:AB_477171 | Dilution Used for IF: 1:200 |
| Antibody | anti-Tuj1 (Mouse monoclonal) | Abcam | Cat#: ab18207 RRID:AB_444319 | Dilution Used for IF: 1:200 |
| Transfected construct (Mouse) | end-binding protein 3 (EB3) -mCherry | Obtained by Dr. Erika Holzbaur | | (*Guedes-Dias et al., 2019*) |
| qRT-PCR primer | *Tubb4a* Primers | Integrated DNA Technologies | Custom Designed | Tubb4a primer: Probe: 5'-/5FAM/ ATGACCTCC/ZEN/ CAGAACTTGGCCC/3IABkFQ /- 3' Primer 1: 5' GACACCCGTCCATCAGCA3' Primer 2: 5'GTCGATGCCGTGCTCAT-3' |
| qRT-PCR primer | *sfrs9* Primers | Integrated DNA Technologies | Custom Designed | Probe: 5'-/5HEX/CAGACATCC/ZEN/ CCAGCTTCTCGCAT/3IABkFQ /- 3' Primer 1: 5' TTCAACCATCCCCATTCCG-3' Primer 2: 5' CCTCCTACAACAAGACGGTCAGAT-3' |
| Software | Graphpad Prism | Graphpad Prism | Graphpad Prism 9 RRID:SCR_002798 | |
| Other | DAPI stain | Invitrogen | Cat#: P36931 | 1 µg/mL |

## Generation of mouse model

Heterozygous $Tubb4a^{D249N/+}$ mice were generated using CRISPR-Cas9 technology by inserting the c.745G > A (p.Asp249Asn) mutation in exon 4 of the mouse *Tubb4a* gene. Methodologies are available in Appendix 1 and gRNA sequences is listed in the key resources table. Sequencing of other areas with high homology to the target sequence was performed to assess that only variants in the *Tubb4a* gene were retained in the final mouse line. Control (WT) mice were generated from breeding pairs of heterozygous $Tubb4a^{D249N/+}$ mice to further provide reassurance of limited of target effects in other homologous genes.

We established one founder line of $Tubb4a^{D249N/+}$ mice, which were bred to produce wild type (WT), $Tubb4a^{D249N/+}$, and $Tubb4a^{D249N/D249N}$ mice. The animals were genotyped at all experimental steps and were maintained under a 12 hr (h) light:12 hr dark cycle in a clean facility and given free access to food and water. The methods and study protocols were approved in full by the Institutional Animal Care and Use Committee of the Children's hospital of Philadelphia and conformed with the revised National Institutes of Health Office of Laboratory Animal Welfare Policy.

## Behavioral analysis

Ambulatory angle, ambulation, hanging grip strength, righting reflex (*Feather-Schussler and Ferguson, 2016*), and rotarod (*Shiotsuki et al., 2010*) were assessed at defined developmental intervals (*Figure 1C*). Detailed methods are provided in SI Methods. A cohort of at least 10 animals per genotype were included for the behavioral tests.

## RNA extraction, cDNA synthesis and qPCR

To determine the relative expression of *Tubb4a* in brain, fresh brain areas as cerebellum, cortex, hippocampus, hypothalamus, pre-frontal cortex, striatum and spinal cord were dissected at end-stage (~P35-P40) and snap frozen. RNA extraction, C-DNA and qPCR details are provided in Appendix 1.

## Tissue processing

Mice were anesthetized based on weight with ketamine and xylazine and transcardially perfused with 4% paraformaldehyde (PFA) after an initial flush with 1X PBS. Brains were collected and post-fixed with 4% PFA in 1X PBS overnight, infiltrated in 30% sucrose in 1X PBS, embedded in optimal cutting temperature compound and sectioned either as coronal or sagittal (50 μm) sections on a cryostat microtome (CM 3050 s, Leica biosystems, USA).

## Immunohistochemistry and image acquisition

Myelin quantification was performed by Eriochrome Cynanine (Eri-C) stain, PLP and MBP in the densely myelinated region- corpus callosum (*Baxi et al., 2017*) and cerebellum. Eriochrome Cyanine (Eri-C) and neurofilament staining was according to previously published protocols (*Sahinkaya et al., 2014*) (Appendix 1). For Nissl staining, frozen sections were stained with 0.1% cresyl violet for 15 min, rinsed with PBS, dehydrated with graded alcohols (from 70–100%), followed by xylene, and mounted with Permount. For immunofluorescence staining, free floating sections were blocked with 2% bovine serum albumin and 0.1% Triton X-100 for 1 hr at room temperature and then sequentially incubated with primary antibodies overnight at 4°C and the fluorescent secondary antibodies for 1 hr at room temperature. Primary antibodies are listed in the Key Resource Table above. AlexaFluor-488 or AlexaFluor-647 conjugated secondary antibodies against rabbit, mouse, or rat (1:1000; Invitrogen) were used, and nuclei were counterstained with DAPI. Image acquisition and image quantification details are provided in the SI.

## Immunoblotting

To determine the relative levels of key myelin proteins in the affected regions of our mouse model and corresponding to areas of disease in H-ABC, immunoblotting was performed on cerebellum and forebrain at post-natal (P)14, P21, and end-stage (~P35-P40). The details of immunoblotting are explained in SI.

## Electron microscopy (EM)

A separate cohort of mice were perfused transcardially with 0.9% saline followed by 2% PFA and 2% glutaraldehyde in 0.1M phosphate buffer (PB; pH 7.4) at end-stage (~day 35) (n = 3/group) (*Lancaster et al., 2018*). The optic nerve and cervical spinal cord were examined as they are highly myelinated and routinely used for studying myelin ultrastructure (*Lancaster et al., 2018*). The optic nerve and cervical spinal cord were dissected, post-fixed for 24 hr, rinsed in 0.1M PB, transferred to 2% OsO4 in 0.1M PB for 1 hr, then processed for embedding in Epon (*Lancaster et al., 2018*). Semi-thin sections were cut, stained with alkaline toluidine blue, and visualized using a light microscope (Lecia DMR) interactive software (Leica Application Suite). Ultra-thin sections (70 nm) were cut, stained with lead citrate and uranyl acetate, and imaged using Jeol-1010 transmission electron microscope. The images for EM sections were assessed using Image J software and the inner and outer axonal area was measured for g-ratio analysis and quantified in 50 axons per animal with n = 3 per group as previously published (*Lancaster et al., 2018*). The axons were classified as normally myelinated, unmyelinated axons (no myelin detected) or thinly myelinated (compared to normal WT tissue) in WT, *Tubb4a*$^{D249N}$ and *Tubb4a*$^{D249N}$ optic nerves (n = 3/group). The percentage of these three groups were plotted and the data was analyzed.

## Oligodendrocyte cultures

Primary oligodendrocyte precursor cells (OPCs) were isolated from cerebral cortices of WT, *Tubb4a*$^{D249N/+}$ and *Tubb4a*$^{D249N/D249N}$ mice between postnatal day P4-P7 using the Miltenyi anti-O4 microbeads (*Flores-Obando et al., 2018*) (Appendix 1). O4+ cells were plated at the density of 20,000 OPCs per well of a 24 well plate, allowed to proliferate for 5–7 days, following which they were differentiated in the media without platelet-derived growth factor and basic fibroblast growth factor and with thyroxine T4 (20 µg/ml) for another 5 days. The cells were then fixed with 4% PFA, washed twice with 1X PBS, permeabilized with 0.2% Triton X-100, blocked in 10% normal goat serum solution for 1 hr, and incubated overnight at 4°C with primary antibodies – rabbit anti-Olig2 (1:800; EMD Millipore, Cat: AB9610), rat anti-PLP (1:1), mouse anti-A2B5 (1:1), rabbit anti-Caspase3 (1:200) and rat anti-MBP (1:1) (IDDRC hybridoma, courtesy Dr. Judith Grinspan). The cells were washed the next day in PBS, and incubated with the appropriate secondary fluorescent antibodies (1:500; anti-rat IgG Alexa Fluor 488, anti-rabbit IgG Alexa Fluor 647; 1:500; anti-mouse IgM Alexa Fluor 488), mounted in Prolong gold antifade reagent (Thermo Fisher Scientific), and imaged using a Nikon microscope with 20 × or 40 × objectives for analysis of cellular counts. Each experiment was repeated at least three separate times and based on the litters, 1–3 pups of the same genotype were used with technical replicate of n = 3/ pup for each experiment.

## Cortical neuron cultures

Primary cortical neurons were isolated from E15.5 embryos of WT, *Tubb4a*$^{D249N/+}$, and *Tubb4a*$^{D249N/D249N}$ mice as described previously (*Guedes-Dias et al., 2019*). In brief, the cortex was dissected from each embryo and washed with Hanks' Balanced Salt Solution (HBSS), then 2.5% trypsin was added to each sample and incubated at 37°C for seven mins. The trypsin was removed and washed four times with fresh warm HBSS and then resuspended in attachment medium (described in SI). The cells were dissociated and plated on poly-L-lysine (PLL) coated MatTek dishes at a density of 150,000 cells/plate for imaging and 100,000 cells/well of a 24 well plate. The medium was changed after 4 hr to pre-equilibrated maintenance media (described in SI). After 3 days, 20–30% of the medium was replaced with fresh medium supplemented with the mitotic inhibitor 1 µM Cytosine arabinoside. The neurons were plated at equal densities for all genotypes in 24-well plate were assessed for cell survival, axonal and dendritic length at 1 week post-plating. Neurons were immunostained with anti-microtubule associated protein (MAP2; 1:200; Sigma, Cat: 1406) and anti- Neuron-specific Class III β-tubulin (TuJ1; 1:200; Abcam, Cat: ab18207), labelled by appropriate secondary fluorescent antibodies (1:500; anti-rat IgG Alexa Fluor 488, anti-rabbit IgG Alexa Fluor 647), and imaged using a Nikon microscope with at 20 × or 40 × objective for cellular counts and measuring axonal and dendritic length using the Neurite tracer plugin in FiJi software. Each experiment was repeated at least three separate times and based on the litters, 1–3 pups of the same genotype were used with technical replicate of n = 3/ pup for each experiment.

## Live-Imaging of EB3 dynamics

On day in vitro (DIV) 6, cortical neurons were transfected with end-binding protein 3 (EB3) -mCherry using Lipofectamine 2000 (Invitrogen). 20–24 hr after transfection, maintenance medium was exchanged to low fluorescence Hibernate E imaging medium (BrainBits) supplemented with 2% B27 and 2 mM GlutaMAX. The neurons were imaged in an environmental chamber at 37°C on a PerkinElmer UltraView Vox Spinning Disk Confocal system with a Nikon Eclipse Ti inverted microscope using a Plan Apochromat 60 × 1.40 NA oil immersion objective. Axons were identified by established morphological criteria and images were acquired with a Hamamatsu EMCCD C9100-50 camera driven by Volocity software (PerkinElmer) at a rate of 2 s per frame for 600 s. Quantification of EB3 dynamics was performed as previously described (*Guedes-Dias et al., 2019*). The ImageJ macro toolset KymoClear was used to generate kymographs (*Mangeol et al., 2016*). The KymoClear toolset passes a Fourier filter on the original kymograph allowing for automated discrimination of anterograde, retrograde or static components and improves the signal-to-noise ratio of EB3 comets without affecting quantitative analysis of the data. The tracks of individual EB3 comets were manually traced using a custom MATLAB GUI (Kymograph Suite) and used to determine run-length, run-time, and velocity of each comet. The investigator was blinded for the neuronal genotype during both image acquisition and kymograph analysis. Each experiment was repeated at least three separate

times and based on the litters, 1–3 pups of the same genotype were used with technical replicate of n = 3/ pup for each experiment.

## Statistical analysis

All graph data are presented as the mean ± standard error mean (SEM). For mouse studies, 'n' represents the number of animals used per experiment unless indicated otherwise. Gait abnormalities, righting reflex, rotarod, and weight assessments were analyzed by two-way ANOVA with repeated measures followed by the post-hoc Tukey test. For grip strength and ambulation, one-way ANOVA with post-hoc Tukey test was performed. Survival was analyzed by the Kaplan-Meier method, and the differences between groups were estimated by the Gehan-Breslow-Wilcoxon test. Comparisons in myelin quantification, NeuN, ASPA, NG2, Olig2, Ki-67 and cleaved caspase three counts and fluorescent intensity were analyzed by ordinary two-way ANOVA with multiple comparisons post-hoc Tukey tests. EM analysis for G-ratio and axon diameter was conducted using one-way ANOVA with the post-hoc Tukey test and the % myelin analysis was conducted with two-way ANOVA followed by post-hoc Tukey test. Neuronal survival, axon and dendritic length, and assessment of OL markers examined in vitro were compared using one-way ANOVA with the post-hoc Tukey test. EB3 dynamics in neurons were analyzed using Kruskal-Wallis test and Dunn's multiple comparisons test. All statistical analyses were performed using Prism 7.0 (GraphPad Software) with p<0.05 considered statistically significant.

# Acknowledgements

We thank Dr. Judith Grinspan and the IDDRC for the availability and use of PLP antibody for the experiments in this manuscript. Additionally, we thank Dr. Kelly Jordan-Sciutto for allowing the use of Keyence microscope for imaging.

# Additional information

## Competing interests

Adeline Vanderver: has research grants support by Gilead, Ionis, Eli Lilly, Illumina and Shire/Takeda, however none of these sources contributed to the current project. The other authors declare that no competing interests exist.

## Funding

| Funder | Grant reference number | Author |
| --- | --- | --- |
| The Commonwealth Universal Research Enhancement | RFA 67-54 | Quasar Padiath Adeline Vanderver |
| National Institutes of Health | R35 GM126950 | Erika LF Holzbaur |
| National Institutes of Health | R37 NS060698 | Erika LF Holzbaur |
| German Research Foundation | Fellowship BO 5434/1-1) | C Alexander Boecker |
| National Institutes of Health | R33NS104384 | Quasar Padiath |
| National Institutes of Health | R33NS106087 | Quasar Padiath |
| National Institutes of Health | R01NS095884 | Quasar Padiath |

The funders had no role in study design, data collection and interpretation, or the decision to submit the work for publication.

## Author contributions

Sunetra Sase, Conceptualization, Resources, Data curation, Software, Formal analysis, Validation, Investigation, Visualization, Methodology; Akshata A Almad, Conceptualization, Data curation, Software, Formal analysis, Supervision, Validation, Investigation, Visualization, Methodology, Project administration; C Alexander Boecker, Pedro Guedes-Dias, Data curation, Investigation,

Methodology; Jian J Li, Resources, Data curation, Investigation, Methodology; Asako Takanohashi, Conceptualization, Data curation, Supervision, Methodology, Project administration; Akshilkumar Patel, Conceptualization, Data curation, Methodology; Tara McCaffrey, Heta Patel, Divya Sirdeshpande, Data curation, Investigation; Julian Curiel, Judy Shih-Hwa Liu, Conceptualization; Quasar Padiath, Conceptualization, Funding acquisition; Erika LF Holzbaur, Resources, Data curation, Formal analysis; Steven S Scherer, Resources, Data curation, Formal analysis, Methodology; Adeline Vanderver, Conceptualization, Resources, Formal analysis, Supervision, Funding acquisition, Validation, Investigation, Visualization, Methodology, Project administration

## Author ORCIDs
Akshata A Almad ⓘ https://orcid.org/0000-0001-6603-2564
Erika LF Holzbaur ⓘ http://orcid.org/0000-0001-5389-4114
Adeline Vanderver ⓘ https://orcid.org/0000-0002-6290-6751

## Ethics
Animal experimentation: This study was performed in strict accordance with the recommendations in the Guide for the Care and Use of Laboratory Animals of the National Institutes of Health. All of the animals were handled according to approved institutional animal care and use committee (IACUC) protocols (17-001250_AM04) of Children's Hospital of Philadelphia. All animals were housed along with littermates and every effort was made to minimize suffering, especially by providing food and hydrogel in the cage of sick mice and conducting euthanasia when they reached endstage.

## Decision letter and Author response
Decision letter https://doi.org/10.7554/eLife.52986.sa1
Author response https://doi.org/10.7554/eLife.52986.sa2

## Additional files
### Supplementary files
• Transparent reporting form

### Data availability
All data generated or analysed during this study are included in the manuscript and supporting files.

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

## Appendix 1

## Supplementary methods

### Generation of mouse model

Heterozygous *Tubb4a*^*D249N* mice were generated using CRISPR - Cas-9 technology by inserting the p.Asp249Asn (c.745G > A) mutation in exon 4 of the *Tubb4a* gene. Cas9 mRNA, gRNA and oligonucleotides (with targeting sequence, flanked by 120 bp homologous combined on both sides) were co-injected into zygotes of C57BL/6 mouse. Following is the gRNA sequence used- gRNA1 (matches forward strand of gene): CAATGCAGATC TACGCAAGCTGG gRNA2 (matches reverse strand of gene): TGCGTAGATCTGCATTGAGC TGG

The resulting CRISPR knock-in mouse model had the heterozygous point mutation of c.745G > A in one allele of the *Tubb4a* gene (*Tubb4a*^*D249N*). Known off-target effects include one synonymous mutation in cis at p.Lys244Lys (**c.732C > A**) with the pathogenic variant at p. Asp249Asn (variant at c.745G > A).

### Mouse genotyping

DNA extraction from tail was carried out by using Hot-SHOT method as published previously (*Truett et al., 2000*). The taq-takara system was used to amplify the 541 bp of PCR product, using the forward primer 5'CCGAGAGGAGTTTCCAGACAGACAGGATC3' and the reverse primer 5'GCTCTGCACACTTAACATCTGCTCG 3'. The products of amplification were subjected to sequencing to identify the genotype of mouse.

### RNA extraction, cDNA and qPCR

RNA extraction was performed using PureLink RNA Mini Kit (Thermo Fisher Scientific, Cat: 12183018A) according to manufacturer's instructions. After treatment with DNAase (Invitrogen), 500 ng of RNA was used for cDNA with SuperScript IV First-Strand Synthesis System (Thermo Fisher Scientific, Cat: 18091200). The mRNA expression levels of *Tubb4a,* and an endogenous housekeeping gene encoding *Splicing factor, arginine/serine-rich 9 (sfrs9)* as a reference, were quantified using real-time PCR analysis (Taqman chemistry) on an Applied Bio System Quanta Flex 7 (Thermo Fisher Scientific, USA). The results were analyzed using the $\Delta\Delta$CT method. For comparison of Tubb4a expression in all brain areas in WT, *Tubb4a*^*D249N/+*, and *Tubb4a*^*D249N/D249N* mice, $\Delta$CT value of spinal cord was used as control value. PCR primers are as follows:

Quantitative Real-Time Reverse Transcription PCR (qRT-PCR) primers:

Tubb4a primer: Probe: 5'-/5FAM/ATGACCTCC/ZEN/CAGAACTTGGCCC/3IABkFQ /- 3'
Primer 1: 5'-GACACCCGTCCATCAGCA-3'
Primer 2: 5' -GTCGATGCCGTGCTCAT-3'
Sfrs9 primer: Probe: 5'-/5HEX/CAGACATCC/ZEN/CCAGCTTCTCGCAT/3IABkFQ /- 3'
Primer 1: %'-TTCAACCATCCCCATTCCG-3'
Primer 2: 5'-CCTCCTACAACAAGACGGTCAGAT-3'

### Behavioral tests

#### Ambulatory angle

Gait abnormalities were determined by measuring the ambulatory/hind limb foot angle. Ambulatory angle was performed with some modifications (*Feather-Schussler and Ferguson, 2016*). The ambulatory angle was measured weekly at post-natal days P7, P14, P21, P28, and P35. Three measurements were performed using data only when the pup was performing a complete walk in a straight line with both feet flat on the ground.

## Ambulation

Ambulatory deficits were detected as per previous publications with some modifications (*Feather-Schussler and Ferguson, 2016*). Ambulatory behavior was assessed at P7, P10, and P14. Based on these strategies of crawling and walking, we examined if transgenic mice attain their crawling/walking skills later than their WT littermates. Mice were scored using a single trial on crawling, gait symmetry, and limb-paw movement during a straight walk (*Figure 1* and *Table 1*). Ambulation scores were given as 0 = no movement, 1 = crawling with asymmetric movement, 2 = crawling with the symmetric movement, and 3 = walking. As illustrated in *Figure 1E*, throughout crawling, the whole hind paw touches the ground as designated by (#) and tail is low or touching the ground. When transitioning from crawling to walking, the head begins to rise. Walking is seen only when the toes of the hind paw touch the ground and the heel is elevated, designated by [##] (*Feather-Schussler and Ferguson, 2016*). Symmetric limb movement was described as hind paws meet front paws during each step, and each step smoothly transitions to the next step. A mouse exhibiting asymmetric limb movement had inconsistent paw placement and transitions from one step to the next are not smooth.

## Hanging Grip strength

Grasping abilities of front and hind paws were determined by performing hanging grip strength. Hanging grip strength was performed as per previous publication with some modifications with details in supplementary text (*Feather-Schussler and Ferguson, 2016*). The trials were repeated three times and average angle was calculated. A 13' x 9.5' metal screen wire mesh was used to perform the hanging grip strength on P14. The protractor was placed in parallel to wire mesh so as to measure the angle at which pups fall. The mouse was placed on the screen and was allowed to adjust to this novel environment for ~10 s. The screen was slowly inverted to 180 degree and approximate angle of the screen when pup falls off was recorded. The trials were repeated three times and average angle was calculated.

## Righting reflex

Righting reflex tests the mice trunk control and motor co-ordination. Righting reflex was carried out as per previous publication with some modifications (*Feather-Schussler and Ferguson, 2016*). The righting reflex test was performed at every week from P7, P14, P21, P28, P35, and then daily when $Tubb4a^{D249N/D249N}$ mice demonstrated motor impairment. Three trials were carried out and a total of 1 min was given for each trial, if needed. The righting reflex test was performed at every week from P7, 14, 21, 28, 35, and then daily when $Tubb4a^{D249N/D249N}$ mice demonstrated motor impairment. The mouse was placed on its back on the bench pad and was held in position for 5 s. Mouse was released and the time taken to return to the flat position was noted. Three trials were carried out and a total of 1 min was given for each trial, if needed.

## Rotarod

Motor coordination, strength, and balance were assessed using a rotarod apparatus (UGO BASILE S.R.L, Gemonio, Italy). The latency to fall from the rotating rod for three test trials after a training period (P21) was recorded in each trial and mean was used for analysis. To evaluate progressive motor loss, the rotarod test was performed on P28 and P35 along with P21, and the mean latency of fall (in seconds) of each age group of mice was calculated and used for statistical analysis. To adapt with the apparatus, on day 1, mice were placed on the cylindrical rod rotating with the constant speed of 5 rpm for 100 s. The next day, mice were placed at an accelerating speed from 5 to 30 rpm for 300 s over three trials/day with an inter-trial interval of approx. 20 min. On day 3, three test trials were performed at an accelerating speed of 5–30 rpm for 300 s.

## Immunohistochemistry, image analysis and quantification

### Myelin quantification and neurofilament staining

Free-floating sections were treated with 10% hydrogen peroxide in methanol for 20 min and blocked for 1 hr with the blocking buffer (4% bovine serum albumin in 1 x PBS with 0.1%

Triton-x-100) followed by 1:500 chicken anti-NF (1:500, Aves, cat: NFH) in the same blocking buffer for overnight at 4°C. Following primary antibody incubation, sections were incubated with biotinylated anti-chicken secondary antibody (1:1000, Aves, cat: B-1005) for 1 hr and developed by Elite Avidin Biotin Conjugate (Vector) and visualized with DAB substrate. Slides were rinsed in tap water, treated with acetone, rinsed in tap water and immersed in eriochrome cyanine (Eri-C) solution for 30 min. The sections were differentiated in 5% Iron Alum, rinsed with tap water and differentiated in borax ferricyanide. Sections were dehydrated, cleared, mounted with permount (Fischer Scientific, USA) and coverslipped. For myelin quantification in corpus callosum and cerebellum (3–4 sections per mouse, n = at least three for P14, P21, and end-stage (~P35-P40)), images were captured in bright field mode in Keyence BZ-X-700 digital microscope. Images captured at 10 × magnification were tiled with the Keyence BZ-X software. The stained area was measured with the Image J software and then related to the total white matter. Antibody information is provided in Key Resource Table.

### NeuN+ Caspase+ and Olig2+ Caspase+ count

For cerebellar and corpus callosum, (3–4 sections per mouse, n = 3–4 for P14, P21, and end-stage), images were captured at 20 × and 40×, respectively, using z-optical sections at 1–2 µm intervals by using Leica DM6000B fluorescence microscope or Olympus laser scanning confocal microscope by using z-stack with 0.5–1 µm optical intervals. Image J software was used to count NeuN+ Caspase+ or Olig2 Caspase cells with DAPI. Analysis was performed blindly and counts are reported as profiles/mm$^2$.

### ASPA+ and Olig2+ NG2+ counts

Sections labeled for ASPA and NG2+/Olig2+ and counterstained with DAPI were used to quantify the total number of ASPA and NG2+/Olig2+ cells in corpus callosum. All images were captured at 40 × oil immersion lens on an Olympus laser scanning confocal microscope by using z-stack with 0.5–1 µm optical intervals. Image J software was used and a standardized sample box (0.01 mm$^2$) was placed in the regions of interest. Positively labeled cells were identified as ASPA+ or NG2+/Olig2+ or Olig2+ cells which were superimposed with the DAPI nuclei. The final counts are reported as profiles/mm$^2$.

### Fluorescent density and area quantification

To quantify fluorescent positive area and density, the images were captured at 10 × magnification using Leica DM6000B fluorescence microscope. Image J software was used, the areas of interest was selected and Integrated area density and grey values were calculated. Following formula was used to calculate fluorescence.

Corrected total fluorescence = Integrated Density – (Area of selected cell X Mean fluorescence of background readings).

## Immunoblotting

To determine the relative levels of key myelin proteins in cerebellum and forebrain at post-natal (P)14, P21, and end-stage (~P35-P40), fresh brain tissues were dissected and lysed in RIPA buffer (Thermo Fischer Scientific, USA) in presence of protease and phosphatase inhibitors (Sigma-Aldrich, USA). Samples were boiled in Laemmli buffer and electrophoresed under reducing conditions on SDS-PAGE gels (4–15% mini-PROTEAN pre-cast gels, Biorad, USA), transferred onto a nitrocellulose membrane (Trans-blot Turbo transfer system, Biorad, USA) by electroblotting, blocked by blocking buffer (1% non-fat milk, Biorad) prepared in 0.05% tween 20 in Tris buffered saline (TBST), and incubated overnight at 4° C with primary antibodies against rat anti-PLP (1:1000, IDDRC hybridoma, courtesy Dr. Judith Grinspan) and rabbit anti-MBP (1:2000, Abcam, Cat: ab40389) diluted in blocking buffer. The membranes were washed five times with TBST, incubated in secondary Horseradish peroxidase (HRP)-conjugated goat anti-rabbit (1:5000, Santa Cruz, Cat: #sc2357) or goat anti-rat antibody (1:5000, ThermoFisher Scientific, Cat: #31470) for 1 hr in blocking buffer, washed in TBST, and developed using standard electrochemiluminescence (ECL) protocols according to manufacturer's instructions (Pierce ECL, ThermoFisher Scientific). Images were scanned and

analyzed by Image J software. For normalization with loading control, mouse anti-actin (1:4000, Chemicon, Cat: MAB1501R), and mouse anti-vinculin (1:2000, Sigma, Cat: 9131) were used after stripping as per manufacturer's instructions (One-minute Western Blot Stripping buffer, GM Biosciences).

## Oligodendrocyte isolation

In brief, cortex was micro-dissected and isolated from each mouse brain, and meninges were removed to avoid contaminating cultures. The cortex was cut into smaller pieces and dissociated using the Neural Dissociation kit (Miltenyl Biotec (P), Cat: 130-092-628) and as per the protocol, each sample was incubated with enzyme mix 1 (Enzyme P and Buffer X) for 15 mins at 37°C. Next, enzyme mix two was added, and the tissue was mechanically dissociated using a fire-polished Pasteur pipette and incubated for 10 mins at 37°C. This process was repeated two more times to obtain a single cell solution which was applied to a 70 µm strainer and centrifuged for 10 mins at 300xg. The cell pellet was resuspended in 90 µl of PBS buffer (pH 7.2) containing 0.5% bovine serum albumin. 10 µl of Anti-O4 Microbeads was added to the cell pellet, mixed, and incubated for 15 mins in the refrigerator. The cells were then washed with 1–2 ml of buffer and centrifuged at 300xg for 10 mins. The supernatant was aspirated and the cells were resuspended in 500 µl of buffer. The MS MACS column was placed in the magnetic field was rinsed with 500 µl of buffer and then the cell suspension was applied to the magnetic column. The flow-through containing unlabeled cells was collected, and the column was rinsed three times with 500 µl of buffer. The column is then removed from the separator and placed on a suitable collection tube, where the appropriate amount of media is applied and immediately flushed by pushing the plunger into the column. This fraction is the O4$^+$ cells was suspended in media containing neurobasal media with 2% B27, 1% penicillin streptomycin, 1% Glutamine, and the growth factors human basic fibroblast growth factor (100 µg/ml; R and D 233-FB/CF), human platelet-derived growth factor -AA (100 µg/ml; Peprotech 100-13A), and human NT3 (100 µg/ml; Peprotech 450–03).

