## [Decision Letter]

**Acceptance summary:**

This work describes the generation and characterization of a novel knock-mouse model for H-ABC leukodystrophy. Of note, this mouse model carries the most common human TUBB4A mutation. This model demonstrates that defects in oligodendrocyte, loss of oligodendrocytes and neurons in specific brain regions, and oligodendrocyte dysfunction in microtubule dynamics underlie H-ABC pathogenesis. This mouse model represents a seminal resource that will lead to increased understanding of this rare leukodystrophy.

**Decision letter after peer review:**

Thank you for submitting your manuscript "TUBB4A mutations result in both glial and neuronal degeneration in an H-ABC leukodystrophy mouse model" for consideration by *eLife*. Your article has been reviewed by three peer reviewers, one of whom is a member of our Board of Reviewing Editors, and the evaluation has been overseen by Huda Zoghbi as Senior Editor.

The reviewers and the Reviewing Editor drafted this decision letter to help you prepare a revised submission.

Summary

The results presented describe the generation and characterization of a novel knock-mouse model for H-ABC leukodystrophy. Importantly, the mouse model harbors the most common human TUBB4A mutation. This model demonstrates that oligodendrocyte maturation defects, loss of oligodendrocytes and neurons in specific brain regions, and dysfunction in microtubule dynamics underlie H-ABC pathogenesis. Overall, the model is well characterized and will provide significant benefit to future studies of pathogenic mechanisms of H-ABC eukodystrophy. Thus, this mouse model represents a significant resource and should lead to increased understanding of this rare leukodystrophy and a platform for discovery of potential therapeutics.

Essential revisions:

1) Provide details of guide RNA and donor selection and sequence. There also should be discussion or data on potential CRISPR off-targets, especially with a single founder animal.

2) Better synchrony, or at least messaging of rationale, for some of the analyses would help significantly. For OLs in particular it's not clear why OL counts are in one region, EM in another, cell death markers in another, and then in vitro OPC isolation in another.

3) The rationale and conclusions of the in vitro studies need clarified. For neurons, the use of fetal cortical neurons seems disconnected from disease pathology. Additionally, it's not clear if the modest difference in mutant neurons numbers originate from differences in neuron initial numbers, plating survival, or actual survival during the assay period. If cell number/density differences exist between genotypes that could impact the other measurements. For in vitro OL studies, because a single time point was used the claim of a defect in differentiation efficiency (vs. early cell death with proliferation remaining OPCs) needs more support.

---

## [Author Response]

Essential revisions:1) Provide details of guide RNA and donor selection and sequence. There also should be discussion or data on potential CRISPR off-targets, especially with a single founder animal.

The gRNA, donor selection and sequence are now provided in the Key resources table. Additional information is also provided in Materials and methods.

Known off-target effects include one synonymous mutation in cis with the pathogenic variant at p.Asp249Asn (variant at c.745G>A). This variant is at c.732C>A (GGC>GGA which both encode lysine) and is now discussed in the Materials and methods section. Specifically, the following text has been added in Materials and methods and Results sections:

“Heterozygous Tubb4a^D249N^ mice were generated using CRISPR – Cas-9 technology by inserting the p.Asp249Asn (c.745G>A) mutation in exon 4 of the Tubb4a gene. Cas9 mRNA, gRNA and oligonucleotides (with targeting sequence, flanked by 120bp homologous combined on both sides) were co-injected into zygotes of C57BL/6 mouse. Following is the gRNA sequence used-

gRNA1 (matches forward strand of gene): CAATGCAGATCTACGCAAGCTGG

gRNA2 (matches reverse strand of gene): TGCGTAGATCTGCATTGAGCTGG

The resulting CRISPR knock-in mouse model had the heterozygous point mutation of c.745G>A in one allele of the Tubb4a gene (Tubb4a^D249N^). Known off-target effects include one synonymous mutation in cis at p.Lys244Lys (c.732C>A) with the pathogenic variant at p.Asp249Asn (variant at c.745G>A).”

“Sequencing of other areas with high homology to the target sequence was performed and only the above sequence variants were retained in the final mouse line. Additionally, the control WT line was generated by breeding heterozygous mice and these are phenotypically normal, adding greater confidence in the specificity of the final mouse line.”

2) Better synchrony, or at least messaging of rationale, for some of the analyses would help significantly. For OLs in particular it's not clear why OL counts are in one region, EM in another, cell death markers in another, and then in vitro OPC isolation in another.

The approaches used are standardized to specific regions. For example, analysis of the corpus callosum is generally used to assess histologic parameters of myelination. Similarly, EM was conducted in myelin rich regions such as the optic nerve or spinal cord as these are routinely examined for studying myelin ultrastructure. OPC isolation was performed as a secondary validation of OL lineage studies, which is typically performed with cerebral cortices. We have now been added previously published references for these established methodologies for clarity. Specifically, details have been added in the Materials and methods sections in the main text under EM, immunohistochemistry and oligodendrocyte culture sections to clarify our approaches. Finally, in some circumstances we used other structures for validation (e.g. the cerebellum) because these regions are known to be affected in H-ABC.

3) The rationale and conclusions of the in vitro studies need clarified. For neurons, the use of fetal cortical neurons seems disconnected from disease pathology. Additionally, it's not clear if the modest difference in mutant neurons numbers originate from differences in neuron initial numbers, plating survival, or actual survival during the assay period. If cell number/density differences exist between genotypes that could impact the other measurements. For in vitro OL studies, because a single time point was used the claim of a defect in differentiation efficiency (vs. early cell death with proliferation remaining OPCs) needs more support.

We agree with the reviewers and have provided additional clarification around these points. We have added references around the use of fetal neurons to generate in vitro models, which is an accepted protocol in the field. We have additionally clarified that neurons were counted prior to plating (at similar densities) and that data that is shown represents actual survival as an endpoint of the assay, and documents 1-week survival across all cohorts. Specifically, the following text has been added in the Materials and methods section: “The neurons were plated at equal densities for all genotypes in 24-well plate were assessed for cell survival, axonal and dendritic length at 1-week post-plating.”

Regarding the OL studies, we have conducted additional experiments to address the question about OPC proliferation and death, in addition to OL maturation and the data are shown in Figure 6—figure supplement 1. We have added the results under the section addressing cell autonomous effects of OPC in vitro studies. We report a significant decrease in proliferation of OPCs isolated from Tubb4a^D249N^ and Tubb4a^D249N/D249N^ mice compared to WT mice, however no significant cell death was observed in the different groups. We further explain in the Discussion, that while the in vitro cell-autonomous effect results in decreased proliferation with no change in cell death; in vivo effects showing increase in proliferation in the context of cell death could be due to cross-talk with other cells such as neurons and influence of the milieu. This has previously been shown in other disorders and a reference is provided.